# NICE 👍 Data Selection for Instruction Tuning in LLMs with Non-differentiable Evaluation Metric

**Jingtan Wang** [1 2]  **Xiaoqiang Lin** [1]  **Rui Qiao** [1 3]  **Pang Wei Koh** [4]  **Chuan-Sheng Foo** [2 5]  **Bryan Kian Hsiang Low** [1 3]

## Abstract

Curating data for instruction tuning is crucial for enhancing the performance of large language models (LLMs). This work aims to select training data for instruction tuning to improve the LLM performance on specific tasks. Existing methods often rely on next-token prediction (NTP) loss as a proxy for target task performance due to the non-differentiable nature of performance evaluation metrics. They select training data points that are most helpful in reducing validation loss. However, there is a discrepancy between minimizing NTP loss and maximizing performance (e.g., code pass rate in code generation). To remedy this, we introduce a novel Non-differentiable evaluation metric-based InfluenCe Estimation (NICE), which leverages the policy gradient to select the training data that improves the performance. Moreover, NICE can perform data selection in the absence of labels (ground-truth responses) when the evaluation metrics do not require labels (e.g., a reward model can output reward scores without supervision from labels). Experimental results show that our approach outperforms existing data selection baselines that use NTP loss in diverse and realistic scenarios. Notably, subsets selected by NICE often produce models that outperform those trained on the full dataset. Our code is available at https://github.com/JTWang2000/NICE.

## 1. Introduction

Instruction tuning (Bai et al., 2022; Ouyang et al., 2022) is a fine-tuning paradigm that enables large language models

(LLMs) to follow specific human instructions, improving their performances on target downstream tasks. The effectiveness of instruction tuning heavily relies on the quality of the instruction dataset (Chen et al., 2023; Li et al., 2024a; Zhou et al., 2024). However, the instruction dataset is usually collected from mixed sources, and some data points may not be directly relevant to the target tasks (Wang et al., 2023; Xia et al., 2024). In addition, the data points often vary in quality and may contain noisy labels (Carlini et al., 2024; Frénay & Verleysen, 2013; Wang et al., 2024a). These challenges underline the importance of data selection methods, which enhance instruction tuning by systematically choosing relevant, high-quality data to cultivate specific target capabilities in LLMs. In practice, LLMs fine-tuned on selected subsets of data can outperform those trained on the full dataset (Wang et al., 2023; Xia et al., 2024).

Loss-based influence estimation methods (Kwon et al., 2024; Xia et al., 2024; Yeh et al., 2022) have been demonstrated to be effective in data selection. It estimates the effect of each training data on the validation loss (e.g., Next-Token Prediction (NTP) loss) via the gradient of the validation loss, then selects the subset of data with the most positive influence. However, many instruction-following tasks require generating long-form responses, which are evaluated using *non-differentiable metrics* (instead of the differentiable validation loss). These evaluation metrics, such as the code pass rate (Chen et al., 2021), LLM-judge (Dubois et al., 2023; Zheng et al., 2023) and reward model (Ouyang et al., 2022), cannot directly provide useful gradient information to estimate the influences due to their non-differentiable nature. Moreover, minimizing NTP loss may poorly align with maximizing the evaluation metrics due to overfitting to surface-level patterns (e.g., n-grams) and ignoring alternative correct generations (Brown et al., 2020; Gloeckle et al., 2024; Tay et al., 2021; Zhou et al., 2024). For instance, in code generation (Chen et al., 2021), there are multiple ways of writing 'correct' code for a problem, but NTP loss is only measured w.r.t one such way. This mismatch between the NTP loss and the true evaluation metric poses a significant challenge to data selection for instruction tuning. Therefore, existing approaches that rely on the influence of NTP loss may fail to select the dataset that improves the metrics used in specific tasks.

[1]National University of Singapore [2]Institute for Infocomm Research (I²R), A*STAR, Singapore [3]Singapore-MIT Alliance for Research and Technology [4]University of Washington [5]Centre for Frontier AI Research (CFAR), A*STAR, Singapore. Correspondence to: Bryan Kian Hsiang Low <lowkh@comp.nus.edu.sg>.

*Proceedings of the 42nd International Conference on Machine Learning*, Vancouver, Canada. PMLR 267, 2025. Copyright 2025 by the author(s).

To tackle this challenge, we introduce a novel Non-differentiable evaluation metric-based InfluenCe Estimation (NICE) method. NICE selects data that directly optimizes commonly used yet non-differentiable evaluation metrics of long-form generation tasks. Inspired by reinforcement learning (RL) (Williams, 1992; Wu et al., 2018; Sutton & Barto, 2018), NICE treats the evaluation metric as the reward function and the LLM as policy. This formulation allows us to overcome the non-differentiability by computing the policy gradient of the metric w.r.t. the model parameters. In particular, the policy gradient is calculated based on the gradients of the likelihood of the model-generated responses, weighted by their corresponding rewards. By using the policy gradients, NICE directly quantifies the influence of training data on validation performance measured by the metric. Therefore, NICE-selected data can better align with the evaluation metrics than the data selected by loss-based influence estimation. Moreover, NICE has two additional advantages: First, NICE supports data selection with unlabeled validation data when the reward function only requires the input and the model-generated response (e.g., the reward model in Bai et al. (2022)), rendering wider applicability and lower annotation costs compared to loss-based influence estimation. Second, NICE is able to use responses generated from better-performing LLMs on the target tasks to further improve the data selection performance as NICE can make use of these high-quality generated responses (instead of only the label used in loss-based influence estimation).

We perform comprehensive analyses to demonstrate the advantages of NICE. First, we empirically show the effectiveness of NICE across diverse and realistic scenarios for instruction tuning. This includes (1) selection from a task-agnostic training data pool that is mixed-source instruction tuning datasets and (2) selection from a task-aware training data pool whose datasets are closely aligned with downstream tasks. Our experiments show that models trained on data subsets selected by our approach generally outperform those trained using either data subsets selected by other baselines or the full dataset. Second, we demonstrate the generality of NICE by applying it to multiple loss-based influence estimation frameworks and empirically verifying their resulting improved performance.

## 2. Preliminaries

Denote an LLM parameterized by $\theta$ as $f(\cdot; \theta)$. Let $x, y$ be the random variables (RVs) for the input (prompt) and the output (response) of the LLM, respectively. Let $y'$ be the RV for a single-token output. Let $D_N := \{z_i = (x_i, y_i)\}_{i=1}^n$ denote the training set where $z_i$ consists of the prompt $x_i$ (a sequence of words or tokens) and the label response $y_i$ (the

ground truth sequence of words or tokens). Similar notations apply to the validation set $D_V := \{z_v = (x_v, y_v)\}_{v=n+1}^{n+m}$. $D_V$ can contain different subtasks: $D_V^1, ..., D_V^q$ [1]. The LLM generates a sequence of words, denoted as $\hat{y}_i = [\hat{y}_i^p]_{p=1}^P$. Here, $\hat{y}_i^p$ is the $p$-th word (or token) in the generated response, and the autoregressive generation process can be described recursively as: $\hat{y}_i^p \sim f(y'|x_i, \hat{y}_i^1, \ldots, \hat{y}_i^{p-1}; \theta)$. The NTP loss for the training data point is defined as

$$L(z_i; \theta) = -P^{-1} \sum_{p=1}^P \log f(y_i^p | x_i, y_i^1, \ldots, y_i^{p-1}; \theta) .$$

The NTP loss for the validation data point is defined likewise. In the rest of the section, we first restate two representative loss-based influence estimation frameworks: TracIn (Pruthi et al., 2020) and Influence Function (Koh & Liang, 2017). Then we review how to use these influence scores to select training data points.

### 2.1. TracIn and Influence Function

**TracIn** quantifies the influence of a training data point $z_i$ on the loss of a validation data point $z_v$ during training. Denote $\eta_t$ as the learning rate used in the parameter update. At each step $t$, the influence is expressed as

$$L(z_v; \theta^{t+1}) - L(z_v; \theta^t) \approx -\eta_t \langle \nabla_\theta L(z_v; \theta^t), \nabla_\theta L(z_i; \theta^t) \rangle,$$

which is the gradient similarity between $z_v$ and $z_i$ derived in App. E.1. To measure the influence of $z_i$ over the entire training run, TracIn aggregates the influence at every training step that uses $z_i$. As $z_i$ is used once per epoch, it is natural to express this as a summation over epochs:

$$\text{Inf}_{\text{TracIn}}(z_i, z_v) = \sum_{e=1}^E \bar{\eta}_e \langle \nabla_\theta L(z_v; \theta^e), \nabla_\theta L(z_i; \theta^e) \rangle$$

where $\bar{\eta}_e$ denotes the average learning rate applied in the $e$-th epoch, $E$ is total number of training epochs, and $\theta^e$ represents the model parameters after the $e$-th epoch.

**Influence Function (IF)** measures the influence of downweighting $z_i$ on the loss of the validation data point $z_v$:

$$\text{Inf}_{\text{IF}}(z_i, z_v) = \nabla_\theta L(z_v; \theta^E)^\top H_{\theta^E}^{-1} \nabla_\theta L(z_i; \theta^E)$$

where $\theta^E$ is the model parameters after the last epoch (total $E$ epochs) and $H_{\theta^E} = \frac{1}{n} \sum_{i=1}^n \nabla_\theta^2 L(z_i; \theta^E)$ is the Hessian matrix of the average training loss over the training set. The derivation can be found in App. E.2.

### 2.2. Targeted Data Selection

The objective of data selection is to identify an optimal subset $D_S \subset D_N$ such that training a model $f$ on $D_S$ achieves comparable or superior performance on downstream tasks

---

[1] In this paper, we examine several datasets, including AlpacaEval, which exhibit this characteristic. Additionally, we investigate datasets that lack explicit subtasks where $q = 1$.

compared to training on the full dataset. It is achieved by selecting training data that maximizes the performance of the *target* task's validation set $D_V$, thereby enhancing model performance on target tasks.

Loss-based influence estimation methods quantify the influence of the individual training data point on validation loss. The influence scores are typically higher for training data which reduces the validation loss more. When loss serves as a proxy for the validation performance, higher scores indicate greater helpfulness for the target task when they are included in the training. To apply influence estimation for data selection, it is necessary to aggregate the scores across the validation set, which may consist of multiple subtasks. Specifically, the influence score of a training data point $z_i$ on each subtask is first computed by averaging the influence scores across the validation data within that subtask. The overall influence for the validation set $D_V$ is then calculated as the maximum influence score across all subtasks:

$$\text{Inf}(z_i, \mathcal{D}_V) = \max_j |\mathcal{D}_V^{(j)}|^{-1} \sum_{z_v \in \mathcal{D}_V^{(j)}} \text{Inf}(z_i, z_v)$$

where $\text{Inf}(z_i, z_v)$ denotes an influence estimation (such as $\text{Inf}_{\text{TracIn}}$ or $\text{Inf}_{\text{IF}}$) that aims to assign higher scores to more helpful training data points. The use of the $\max$ function ensures that training data improving performance on at least one validation subtask are prioritized (Xia et al., 2024). Based on these scores, the top-ranked training data points are selected to construct the training subset $D_S$. This subset is then used to fine-tune the target model.

## 3. Methodology

### 3.1. Non-differentiable evaluation metric-based InfluenCe Estimation (NICE)

Loss-based influence estimation methods quantify the effect of a training data point on the validation loss (e.g., NTP loss), which is a differentiable proxy for the validation performance. However, there are two major drawbacks of loss-based influence estimation: (a) There is a discrepancy between the NTP loss and the evaluation metrics of instruction-following tasks, especially those that require long-form generations (e.g., LLM-judge (Dubois et al., 2023) and code generation benchmarks (Chen et al., 2021)). In other words, selecting training data that minimizes NTP loss on validation data does not necessarily improve the performance for these tasks (as shown in Fig. 1). (b) While obtaining the prompt of a validation data point $x_v$ is relatively easy, the high-quality label $y_v$ may not always be available.

We propose to directly compute the influence of each training data point on the evaluation metric instead of the loss. Specifically, denote the reward function as $r$ (defined by an evaluation metric), which calculates the model performance

as follows:

$$r(z_v, \hat{y}_v) := \begin{cases} r(x_v, y_v, \hat{y}_v) & \text{when } y_v \text{ is required, e.g.,} \\ & \quad \text{LLM judge in (Dubois et al., 2024);} \\ r(x_v, \hat{y}_v) & \text{when } y_v \text{ is not required, e.g.,} \\ & \quad \text{reward model in (Bai et al., 2022).} \end{cases}$$

Note that the ground truth response $y_v$ is not always required by the reward function, depending on the evaluation metric used here. We will use the terms "reward function" and "evaluation metric" interchangeably in the rest of our paper.

To calculate the influence of a training data point on a non-differentiable $r$, we cannot directly apply the same formula as the loss-based influence estimation such as TracIn or IF, because they require the gradient of $r(z_v, \hat{y}_v)$ w.r.t. the model parameters, which is not available. To address this, we propose to use the policy gradient from RL (Wu et al., 2018; Sutton & Barto, 2018). Specifically, we adopt the RL objective function for a validation data point:[2]

$$L_r(z_v; \theta) = \mathbb{E}_{\hat{y}_v \sim f(y|x_v; \theta)}[-r(z_v, \hat{y}_v)]$$

where $f(\cdot; \theta)$ denotes the policy defined by the LLM with parameter $\theta$, which is used to generate response $\hat{y}_v$ for $x_v$. Subsequently, the policy gradient of $L_r$ w.r.t. the model parameters can be derived using the log derivative trick (Williams, 1992; Meyer, 2023):

$$\nabla_\theta L_r(z_v; \theta) = \mathbb{E}_{\hat{y}_v \sim f(y|x_v; \theta)}[-\nabla_\theta \log(f(\hat{y}_v|x_v; \theta))r(z_v, \hat{y}_v)].$$

This can be estimated using Monte-Carlo sampling on the responses generated by $f(x_v; \theta)$, a technique also known as the Monte-Carlo (MC) policy gradient. Intuitively, policy gradient optimizes the model by increasing the probability of generating responses with high and positive rewards. By using the policy gradient, the influence of a training data point $z_i$ on the model performance on a validation data point $z_v$ measured by the reward function $r$ at time step $t$ is calculated as

$$L_r(z_v; \theta^{t+1}) - L_r(z_v; \theta^t) \approx -\eta_t \langle \nabla_\theta L(z_i; \theta^t), \nabla_\theta L_r(z_v; \theta^t) \rangle.$$

The approximation above is derived using a similar logic as Eq. 1 in App. E that uses the first-order Taylor approximation. Then, we can measure the non-differentiable evaluation metric-based influence of $z_i$ on $z_v$'s performance over the entire training run as

$$\text{Inf}_{\text{NICE}}(z_i, z_v) = \sum_{e=1}^{E} \bar{\eta}_e \Big\langle \nabla_\theta L(z_i; \theta^e),$$
$$\mathbb{E}_{\hat{y}_v \sim f(y|x_v; \theta^e)}[-\nabla_\theta \log(f(\hat{y}_v|x_v; \theta^e))r(z_v, \hat{y}_v)] \Big\rangle.$$

---

[2]We use a negative sign in front of $r$ to make the notations in the influence estimation in the rest of our paper consistent with loss-based influence estimation, i.e., the lower the $L_r(z_v; \theta)$, the better the model (consistent with validation loss).

To interpret, NICE assigns a higher influence score to a training data point $z_i$ if its gradient (of the training loss) is more similar to the policy gradients of the validation performance evaluated by the reward function $r$. The higher the influence score, the larger the performance measured by the evaluation metric improves when including training data point $z_i$. To apply NICE for data selection, the same aggregation of the influence scores in Sec. 2.2 is applied. We then select the data subset $D_S$ by including the training data points with top-ranked aggregated influence scores.

To summarize, NICE enables data selection to directly optimize the non-differentiable evaluation metrics via influence estimation using policy gradient. It also enables data selection with unlabeled validation data when the metric does not require the label as input.

Although there are various alternatives to compute the gradient of $L_r$ from policy optimization research (Schulman et al., 2017; Rafailov et al., 2023), we use the MC policy gradient as it is easy to implement and has been shown effective in many applications. We further demonstrate in App. I.1 that gradients computed by other policy optimization approaches can also be used in NICE to achieve better performance than loss-based influence estimation.

*Remark* 3.1 (Equivalence to TracIn). When the label response consists of a single token and the evaluation metric is accuracy, NICE is equivalent to TracIn. Thus, for tasks that do not require generating long responses, vanilla loss-based influence estimation performs similarly to NICE.

*Remark* 3.2 (Empirical consideration). To improve the performance of NICE, we integrate the two improvements proposed by LESS (Xia et al., 2024), which adapts TracIn for influence estimation on LLM. Specifically, we use the Adam gradient for training data instead of the SGD gradient and replace the inner product with cosine similarity (i.e., equivalent to normalizing the gradient before the inner product) in the definition of influence to mitigate the bias toward short sequences. The explicit form of NICE used in our implementation is elaborated in App. F.

### 3.2. Generalization to Other Loss-based Influence Estimation Methods

We have discussed the use of policy gradient on a specific loss-based influence estimation method – TracIn – to estimate the influence of data points on the non-differentiable evaluation metrics. However, our approach is not limited to TracIn alone. The policy gradient can be applied to other methods, such as the influence function:

$$\text{Inf}_{\text{NICEIF}}(z_i, z_v) = \nabla_\theta L_r(z_v; \theta^E)^\top H_{\theta^E}^{-1} \nabla_\theta L(z_i; \theta^E)$$

where $E$ is the number of epochs and $\nabla_\theta L_r(z_v; \theta^E) = \mathbb{E}_{\hat{y}_v \sim f(y|x_v; \theta^E)}[-\nabla_\theta \log(f(\hat{y}_v|x_v; \theta^E))r(z_v, \hat{y}_v)]$. The derivation follows from a similar logic as Eq. 2 in App. E

by first quantifying the influence of the training data on the parameter, then using the chain rule to calculate the impact of this influence on $L_r$ at the validation data. A higher $\text{Inf}_{\text{NICEIF}}$ means a larger increase in $L_r$ and hence a larger decrease in performance measured by $r$ when downweighting the training data (i.e., removing the data point makes the model perform worse), indicating a higher quality of that training data point. The same aggregation method described in Sec. 2.2 is used for aggregating $\text{Inf}_{\text{NICEIF}}$.

Our implementation uses a similar approach as DataInf to improve the efficiency of NICEIF by using the first-order derivatives to estimate the Hessian inverse, which is required in the calculation of the IF (Kwon et al., 2024).

### 3.3. Assisted Monte-Carlo Sampling

We use MC sampling to estimate the policy gradient used in NICE. Recall that NICE uses the MC policy gradient. Specifically, for a prompt $x_v$ of a validation data point, we sample multiple responses from the LLM and use the sample mean to estimate the policy gradient. There are two major advantages of using this MC policy gradient compared to the gradient used in loss-based influence estimation: 1) Policy gradient estimated using MC utilizes multiple different responses, offering diverse guidance compared to the label response; 2) The generated response can be better than the label response (as demonstrated in Tab. 1 where the label response is incorrect and less detailed) and hence result in better data selection performance.

Despite these advantages, the MC policy gradient has its practical limitations. Specifically, when the model (policy) is too weak for the task, the MC samples may not contain high-quality responses with high rewards. As a result, the corresponding estimated policy gradient will not contain signals for improving the policy's performance. To ensure the quality of the generated responses, we propose an alternative approach named assisted Monte-Carlo (AMC) sampling, which uses a model $g(\cdot; \psi)$ parameterized by $\psi$ that is better at the target task to assist the response generation:

$$\text{Inf}_{\text{NICEAMC}}(z_i, z_v) = \sum_{e=1}^{E} \bar{\eta}_e \Big\langle \nabla_\theta L(z_i; \theta^e),$$
$$\mathbb{E}_{\hat{y}_v^* \sim g(y|x_v; \psi)} \big[-\nabla_\theta \log(f(\hat{y}_v^*|x_v; \theta^e)) \, r(z_v, \hat{y}_v^*)\big] \Big\rangle.$$

The generated responses $\hat{y}_v^* \sim g(x_v; \psi)$ with the assistance from better-performing LLMs have better quality assurance than vanilla MC sampling (Tab. 1). With AMC sampling, NICEAMC is more likely to select a data subset that guides the model to generate better responses, improving the performance.

## 4. Experiments

We experimentally analyze model performance under two different procedures for forming the **initial training data**

*Table 1.* Generated responses from different models for a validation data point in AlpacaEval task. Llama2-7B or Mistral-7 is the model $f$ used in NICE. GPT4 is the 'better-performing' LLM used in NICE$_{AMC}$. As the model improves, its generated response becomes more detailed and can be better than the label response (the label response is incorrect while Mistral-7B and GPT's responses are correct).

| |
|---|
| **Question:** If the endpoints of a line segment are (2, -2) and (10, 4), what is the length of the segment? |
| **Label Response :** The length of the line segment is 12 units. |
| **Llama2-7B Generated Response :** The length of the segment is 10 - (-2) = 12 units. |
| **Mistral-7B Generated Response :** The length is the distance between the endpoints. To find this distance, use the following formula: $(final\_x - init\_x)^2 + (final\_y - init\_y)^2 = distance^2$ 
 In this case, the values are $(10, 4)$ and $(2, -2)$. The length is $sqrt(8^2 + 6^2) = sqrt(100) = 10$. |
| **GPT4 Generated Response:** To find the length of a line segment given its endpoints, you can use the distance formula. The distance formula between two points $(x_1, y_1)$ and $(x_2, y_2)$ in the coordinate plane is: $d = \sqrt{(x_2 - x_1)^2 + (y_2 - y_1)^2}$. For the points given, $(x_1, y_1) = (2, -2)$ and $(x_2, y_2) = (10, 4)$. Plugging these into the formula: $d = \sqrt{(10-2)^2 + (4-(-2))^2}$   $d = \sqrt{8^2 + 6^2}$   $d = \sqrt{64 + 36}$   $d = \sqrt{100}$   $d = 10$. Therefore, the length of the line segment is 10 units. |

**pool**: task-agnostic and task-aware. We evaluate our method on tasks that require long-form generation and use evaluation metrics that are commonly used to evaluate the corresponding tasks. In this section, we describe the evaluation tasks (i.e., the task that we wish to improve on via instruction tuning) in Sec. 4.1, experimental settings in Sec. 4.2, experimental details in Sec. 4.3, main results in Sec. 4.4, and additional analysis in Sec. 4.5.

### 4.1. Evaluation Tasks

We use four evaluation tasks, including AlpacaEval (Dubois et al., 2024), TLDR (Stiennon et al., 2020), RLHF (Bai et al., 2022), and HumanEval (Chen et al., 2021).

*Table 2.* Detailed information about the evaluation datasets.

| Dataset | $|D_V|$ | $|D_{\text{test}}|$ | Evaluation metrics |
|---|---|---|---|
| AlpacaEval | 10 | 795 | length-controlled win rate |
| TLDR | 322 | 6553 | reward model |
| RLHF | 2192 | 2354 | reward model |
| HumanEval | 10 | 154 | pass@$k$ |

**AlpacaEval** is a compilation of prompt-response pairs aimed at assessing language models' instruction following capability. We use the length-controlled win rate to ensure a fair evaluation (Dubois et al., 2024). **HumanEval** evaluates code generation from natural language instructions using the *pass@k* metric, which is the probability of having at least one correct solution (pass a specified unit test) when sampling $k$ responses from the model. Pass@$k$ with $k > 1$ is practical in real-world scenarios when expected behaviors are known and test cases are readily available. It evaluates the test-time scaling capability of the model by allowing multiple candidate solutions ($k$) to be generated and validated to effectively identify the correct code. We adopt pass@100 because it achieves substantially higher accuracy than smaller $k$ (Chen et al., 2021), making it more practical for real-world scenarios. For completeness, we also provide detailed results for different $k$ in App. H.1. **TLDR** con-

tains polished text summaries. The evaluation metric is the reward model (OpenAssistant, 2023) trained on human feedback, measuring the the quality of summaries and alignment with human preference. **RLHF** consists of prompt-response pairs where each includes a "chosen" response that aligns better with human preferences (we use only the "chosen" columns). We use a trained reward model (Ray2333, 2024) as the evaluation metric to evaluate the helpfulness of the responses. For the dataset splits, we randomly select 10 examples from both AlpacaEval and HumanEval as the validation set, with the remainder as the test set. For RLHF, we sample 5% from the training dataset as the validation set since the original dataset only contains train and test splits. For TLDR, we use 5% of the original validation set, given that the original validation set is too large. We study the robustness of our method to different validation splits in App. H.2. We provide a summary of these evaluation tasks in Tab. 2 and additional details on these tasks in App. B.1. Unless specified, the results reported are evaluated on the test set.

### 4.2. Experimental Settings

Our problem setup focuses on targeted data selection, assuming access to validation data during data selection. We further consider two distinct settings where we perform selection from "a task-agnostic training data pool" and "a task-aware training data pool", distinguished by whether the knowledge of the downstream task is available when forming the initial training data pool before data selection. Specifically:

**Selection from a task-agnostic training data pool**. In this setting, a large, diverse, mixed-source pool of instruction tuning training set is collected without the knowledge of the downstream task before data selection. We use Tulu (Wang et al., 2023) as the training dataset, which consists of Chain of Thought (COT) (Wei et al., 2022), Databricks Dolly (DOLLY) (Conover et al., 2023), Open Assistant 1 (OASST) (Köpf et al., 2023), and FLAN V2 (Longpre et al.,

2023). Intuitively, this mixed-source pool of data may contain irrelevant data (e.g., assistant-style conversations) w.r.t. the targeted task (e.g., coding task). Additional details for the training set are in App. A.1.

**Selection from a task-aware training data pool**. In this setting, the training set is collected specifically for the downstream evaluation task, hence being task-aware. We consider two evaluation tasks here: **RLHF** and **HumanEval**, with the evaluation datasets the same as in Sec. 4.1. For RLHF, the training data is 95% of the original training set used for the helpfulness assistant, as provided in Bai et al. (2022). For HumanEval, we adopt the CodeAlpaca 20k (Chaudhary, 2023) dataset as the training set, which is a crowd-sourced collection of code-related instruction-response pairs, designed to fine-tune language models for better performance in code generation and understanding. Additional training set details are in App. A.1.

### 4.3. Experimental Details

**Efficient Data Selection for LLM**. To improve the efficiency of data selection, we train models with LoRA (Low-Rank Adaptation). We adopt the warmup training which trains the LLM on a randomly selected subset of training data for influence estimation and the number of warmup epochs is the $E$. Additionally, random projections are applied to the LoRA gradients, preserving the essential inner products while reducing the dimensionality of the gradient to reduce the memory requirement (Johnson, 1984).

**Models and Hyperparameters**. Our primary evaluation of NICE focuses on two LLMs: Llama2-7B (Touvron et al., 2023), and Mistral-7B (Jiang et al., 2023), with performance averaged over three seeds. Larger and state-of-the-art models, including Llama2-13B and Llama3-8B, were also tested on the RLHF task, presented in App. H.3. We perform warmup training for 4 epochs on 5% of the training set for the task-agnostic training data pool, and 20% for the task-aware training data pool due to its smaller training set size. We project the LoRA gradient into an 8192-dimensional vector. The influence estimates for data points are obtained by the respective data selection approaches, with the top 5% data points (ranked by influence) to be the selected data subset $D_S$ for the task-agnostic setting (20% for the task-aware setting). Multinomial sampling (Chatterjee & Cancedda, 2010) is used to generate the MC samples for NICE. We generate 20 MC samples for all evaluation tasks, except for HumanEval where we generate 500 samples due to the difficulty of the task (i.e., responses having low code pass rates). For NICEAMC, we use GPT-4 as $g(\cdot, \psi)$. Note that when the reward function does not require labels, the ground truth label is not used by our approach, detailed in App. C.

**Baselines**. We evaluate NICE against a variety of baselines. The most straightforward baseline is Random where data points are randomly sampled from the training set for instruction tuning. We also employ BM25 (Robertson et al., 2009), which ranks training data based on relevance to the validation data, and then selects the top-ranked data points to form $D_S$. Another baseline, DSIR (Xie et al., 2023), selects $D_S$ based on n-gram lexical feature matching between training and validation distributions. Representation-based Data Selection (RDS) (Zhang et al., 2018) ranks the training instances using the cosine similarity of the features between the training and the validation data, and we adopt SentenceBERT (Reimers, 2019) embedding as the features. LESS (Xia et al., 2024) uses loss-based influence estimation to select the training data with top influence scores. TSDS (Liu et al., 2024b) also leverages loss-based gradient features, and further optimizes for distribution alignment and diversity via optimal transport and kernel density estimation, respectively. Note that all baselines, except random, are calculated using complete data points (i.e., the concatenation of prompt and response). More implementation details about the baselines are in App. D.

### 4.4. Results

The results of NICE and NICEAMC against baselines are presented in Tab. 3. We summarize the findings below.

**NICE Outperforms Loss-based Influence Estimation**. Our results show that NICE consistently outperforms LESS, which uses loss-based influence estimation for data selection, across various models, settings, and datasets. This result verifies that using the estimated influence on the evaluation metric is more helpful for data selection than that on the loss.

**No Labels? No Problem! NICE Outperforms Baselines that Utilize the Label Response**. For tasks like TLDR, RLHF, and HumanEval, NICE or NICEAMC uses only unlabeled validation data (i.e., only prompts). Surprisingly, they outperform baselines that use labeled data (both prompts and label responses).

**Less Is More: Subset Outperforms the Full Dataset**. We find that the subset selected by NICE or NICEAMC can outperform the full dataset, demonstrating the value of carefully curated data over larger, less refined datasets.

**Assisted Monte-Carlo Sampling Can Boost Data Selection**. Adopting AMC sampling in policy gradient has the potential to further improve the performance of NICE, especially when the initial pool of training data is large. Responses generated from better-performing models can effectively guide data selection, enabling models trained on the selected subset to achieve better performance.

However, NICEAMC does not always have improved performance compared to NICE, particularly under a smaller selection pool of training data (e.g., a task-aware training

*Table 3.* Comparison of NICE and NICE_AMC in both *task-agnostic* and *task-aware* settings for Llama2-7B and Mistral-7B. **Bold** numbers indicate the top-performing selected subset. A purple cell suggests that NICE outperforms LESS which uses loss-based influence estimation. Underlined numbers show that the subset selected by our approach exceeds the performance of the full dataset. Subscript numbers represent standard deviations.

| Model & Dataset | | Full | Random | RDS | BM25 | DSIR | TSDS | LESS | NICE | NICE_AMC |
|---|---|---|---|---|---|---|---|---|---|---|
| *Task-agnostic training data pool.* | | | | | | | | | | |
| Llama2-7B | AlpacaEval | 22.59 | $16.13_{\pm 1.18}$ | 14.70 | 19.60 | 20.27 | $17.40_{\pm 2.44}$ | $26.94_{\pm 2.37}$ | $\underline{27.61}_{\pm 2.13}$ | $\mathbf{\underline{30.45}}_{\pm 2.40}$ |
| | TLDR | 2.40 | $1.80_{\pm 0.08}$ | 2.08 | 2.15 | 1.53 | $2.19_{\pm 0.29}$ | $3.37_{\pm 0.29}$ | $\mathbf{\underline{3.61}}_{\pm 0.78}$ | $\underline{3.55}_{\pm 0.40}$ |
| | RLHF | 2.31 | $2.05_{\pm 0.11}$ | 1.87 | 2.83 | 2.57 | $1.01_{\pm 0.12}$ | $1.44_{\pm 0.07}$ | $\underline{2.82}_{\pm 0.10}$ | $\mathbf{\underline{3.03}}_{\pm 0.02}$ |
| | HumanEval | 47.44 | $44.30_{\pm 2.36}$ | 45.29 | 46.19 | 42.22 | $43.68_{\pm 1.82}$ | $47.50_{\pm 1.57}$ | $\mathbf{\underline{48.59}}_{\pm 2.08}$ | $45.10_{\pm 2.84}$ |
| Mistral-7B | AlpacaEval | 33.77 | $24.99_{\pm 4.28}$ | 21.70 | 28.47 | 29.31 | $35.84_{\pm 0.53}$ | $41.09_{\pm 1.56}$ | $\underline{41.43}_{\pm 3.00}$ | $\mathbf{\underline{47.40}}_{\pm 2.94}$ |
| | TLDR | 2.79 | $3.06_{\pm 0.24}$ | 2.90 | 2.41 | 3.48 | $3.28_{\pm 0.41}$ | $4.40_{\pm 0.12}$ | $\mathbf{\underline{4.80}}_{\pm 0.12}$ | $\underline{4.59}_{\pm 0.20}$ |
| | RLHF | 2.56 | $2.13_{\pm 0.04}$ | 1.78 | 2.88 | 2.94 | $1.83_{\pm 0.15}$ | $1.70_{\pm 0.09}$ | $\underline{3.10}_{\pm 0.06}$ | $\mathbf{\underline{3.42}}_{\pm 0.05}$ |
| | HumanEval | 83.63 | $85.56_{\pm 1.27}$ | 84.15 | 84.09 | 79.17 | $82.78_{\pm 1.25}$ | $85.24_{\pm 0.45}$ | $\underline{85.59}_{\pm 1.41}$ | $\mathbf{\underline{85.67}}_{\pm 0.34}$ |
| *Task-aware training data pool.* | | | | | | | | | | |
| Llama2-7B | RLHF | 1.01 | $1.04_{\pm 0.04}$ | 0.66 | 1.29 | 1.43 | $0.97_{\pm 0.02}$ | $1.62_{\pm 0.05}$ | $\mathbf{\underline{1.69}}_{\pm 0.05}$ | $\underline{1.32}_{\pm 0.05}$ |
| | HumanEval | 51.27 | $51.91_{\pm 1.61}$ | 54.74 | 52.23 | 53.10 | $49.85_{\pm 3.17}$ | $52.67_{\pm 0.71}$ | $\mathbf{\underline{55.09}}_{\pm 1.66}$ | $50.67_{\pm 1.24}$ |
| Mistral-7B | RLHF | 0.99 | $1.05_{\pm 0.04}$ | 0.56 | 1.31 | 1.31 | $1.15_{\pm 0.06}$ | $1.29_{\pm 0.13}$ | $\mathbf{\underline{1.71}}_{\pm 0.01}$ | $\underline{1.35}_{\pm 0.07}$ |
| | HumanEval | 84.27 | $83.34_{\pm 2.54}$ | 86.75 | 84.81 | 79.91 | $85.51_{\pm 1.28}$ | $85.26_{\pm 1.13}$ | $\mathbf{\underline{87.35}}_{\pm 1.03}$ | $84.18_{\pm 1.63}$ |

data pool is often smaller). When the training set is small, there may not be enough training data points with gradients close to the policy gradient of NICE_AMC. We perform a simple experiment to verify this intuition: restricting NICE_AMC to compute score from the RLHF training set yields a performance of 1.26, but expanding the selection pool to the combination of RLHF set and a large instruction tuning set (COT, DOLLY, OASST, Flan V2) increases the performance to 3.35. Note that the additional controlled experiment only expanded the selection pool without altering the initial warmup process. In contrast, the performance of NICE improves from 1.68 to 2.44 with a larger pool. This comparison shows a clear advantage of NICE_AMC when the training data to select from is large in size. Consequently, when the size of training data is large (e.g., a task-agnostic training data pool), we can prioritize using NICE_AMC.

### 4.5. Additional Analysis

Unless specified, the experiments in the section below are conducted on the Llama2-7B model.

**Time Complexity Analysis**. We provide a comparative analysis of the computational costs between NICE and LESS, an approach that adopts loss-based influence estimation, showing that NICE remains within a practical computational range. Tab. 4 lists the asymptotic complexity and wall-clock runtime (the time for warmup training with LoRA is measured in single H100 GPU hours, others are measured in single L40 GPU hours) for each stage in the data selection procedure. Tab. 5 highlights the validation gradient computation where NICE differs from LESS. Let $E$ denote the number of epochs (saved checkpoints), $d$ the

*Table 4.* Comparison of asymptotic time complexity and wall-clock time (in GPU hours) for each stage in data selection. The time for Warmup training with LoRA is measured on H100, and the others are measured on L40.

| | Remark | Asymptotic | Compute |
|---|---|---|---|
| Warmup Training | NICE = LESS | $O(|D_W|E)$ | 3h |
| Training Grad | NICE = LESS | $O(|D_N|E)$ | 48h |
| Validation Grad | NICE > LESS | LESS: $O(|D_V|E)$; NICE: $O(|D_V|EM)$ | LESS: 0.11h on avg; NICE: 14.67h on avg |
| Data Selection | NICE = LESS | $O(|D_N||D_V|d)$ | <0.02h |

*Table 5.* Validation gradient computation time across tasks for NICE and LESS in single L40 GPU hours.

| Task | $M$ | NICE (MC Sampling) | NICE (Val Grad) | LESS (Val Grad) |
|---|---|---|---|---|
| AlpacaEval ($|D_V| = 10$) | 20 | 0.17h | 0.05h | <0.02h |
| TLDR ($|D_V| = 322$) | 20 | 8h | 1.47h | 0.08h |
| RLHF ($|D_V| = 2192$) | 20 | 32h | 10h | 0.33h |
| HumanEval ($|D_V| = 10$) | 500 | 5h | 2h | <0.02h |

dimension of the projected gradients, and $M$ the number of Monte Carlo (MC) samples. Let $|D_W|$, $|D_N|$, and $|D_V|$ denote the warmup, training, and validation set sizes, respectively. When $|D_V|$ and $M$ are small, NICE adds only marginal overhead to LESS (e.g., AlpacaEval).

While NICE utilizing Monte Carlo sampling can indeed increase the computational cost, this trade-off is justified by our approach not needing validation labels—a key motivation of our work. NICE fills a gap left by existing loss-based baselines by supporting data selection with unlabeled validation data in cases where the evaluation metrics are label-independent. Furthermore, we can observe the performance improvement over other methods in Tab. 3.

**Generalizing NICE to the Influence Function (IF)**. We demonstrate the effectiveness of the main idea of NICE beyond the TracIn framework by extending it to IF. We adopt DataInf as an efficient implementation for IF. We compute the DataInf on the last checkpoint of the warmup model and adopt random projection to reduce the dimensionality of gradients and store the projected gradients, eliminating the need for computing the gradients again after the computation of the Hessian. More details on the implementation of DataInf are provided in the App. E.3. We compared the performance of selecting data with the vanilla (loss-based) DataInf and the DataInf enhanced by the policy gradient: NICEIF when the training data pool is task-agnostic (Sec. 4.2). As shown in Tab. 6, NICEIF consistently outperforms DataInf, demonstrating the effectiveness of our approach and showing that the concept of NICE of applying policy gradient for influence estimation can be readily applied to other loss-based influence estimation methods to improve the performance.

*Table 6.* Comparison between Influence Function (DataInf) and NICEIF on Llama2-7B when the training data pool is *task-agnostic*. NICEIF consistently outperforms the DataInf.

| Method | AlpacaEval | TLDR | RLHF | HumanEval |
|---|---|---|---|---|
| IF (DataInf) | 11.11 | 2.01 | 0.83 | 37.40 |
| NICEIF | **20.44** | **3.97** | **1.89** | **39.68** |

**The Discrepancy Between NTP Loss and Evaluation Metrics**. Previous works have discussed the discrepancy between validation loss and downstream performance in instruction tuning for LLMs (Tay et al., 2021; Xia et al., 2024). In these scenarios, minimizing validation loss does not necessarily correspond to improving validation performance, especially when the task requires long-form generations. Empirical observations described in Fig. 1 and further results in App. H.4 verify this discrepancy: The minimized validation loss is achieved at step around 250. However, that checkpoint is the worst-performing checkpoint (lowest validation reward). The reward can be further increased in later steps, even if the loss increases.

We additionally plot the validation performance, measured by the reward of the last few checkpoints, for models trained on NICE-selected subsets and LESS-selected subsets in Fig. 2 for the RLHF task. NICE-selected data optimizes in the direction of increasing validation performance, whereas LESS-selected data prioritizes loss reduction, which may not necessarily lead to improved validation performance.

**The Effect of the Number of Monte-Carlo Samples**. We perform additional experiments to study the effect of the number of Monte-Carlo (MC) samples used in approximating policy gradient on the data selection performance (measured by reward for the RLHF task). The results in Fig. 3 indicate a positive correlation between performance

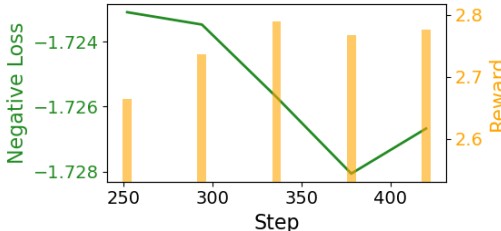

*Figure 1.* Discrepancy between NTP loss and performance (i.e., measured by the reward model here) of the validation set for RLHF task in the last few training steps. The checkpoint with the minimized loss (highest negative loss) corresponds to the checkpoint with a relatively worse performance (having the lowest reward among the five checkpoints). The performance may continue to increase even if the loss increases (negative loss decreases).

and the number of MC samples, which shows the potential to further improve NICE by using more MC samples. However, generating additional MC samples is computationally expensive. We use 20 MC samples for the majority of tasks since it is relatively less computationally expensive, while sufficient to achieve better performance than other baselines. The number of MC samples can also affect the stability of our approach, as elaborated in App. H.5.

**Data Addition**. In Fig. 4, we plot the performance (measured by reward) against the percent of data points selected by NICE and Random for RLHF task. The performance of the model trained on a randomly selected subset increases gradually as more data is used. In contrast, for the NICE-selected subset in the task-aware setting (right of Fig. 4), performance rises slightly from 5% to 25% but drops sharply beyond 25%. The task-aware training data pool is more relevant to the downstream task, so a small percentage can miss useful data, while a large percentage can include irrelevant or harmful data, harming performance. When the training data pool is task-agnostic, performance declines as the selection percentage increases, likely because only a small fraction (no more than 5%) is relevant. This experiment also demonstrates the importance of data selection to exclude data points that are not useful to model performance.

## 5. Related Works

Various approaches are proposed to estimate the influence of training data in fine-tuning LLMs. LESS (Xia et al., 2024) adapts the TracIn framework to estimate the influence of data points in instruction tuning. Kwon et al. (2024) and Choe et al. (2024) scale up the IF by speeding up the computation of the Hessian inverse. Lin et al. (2024) studies the token-level influence function for LLMs. These approaches above are all loss-based influence estimation methods that aim to approximate the influence of data on the validation loss. Consequently, they fall short for genera-

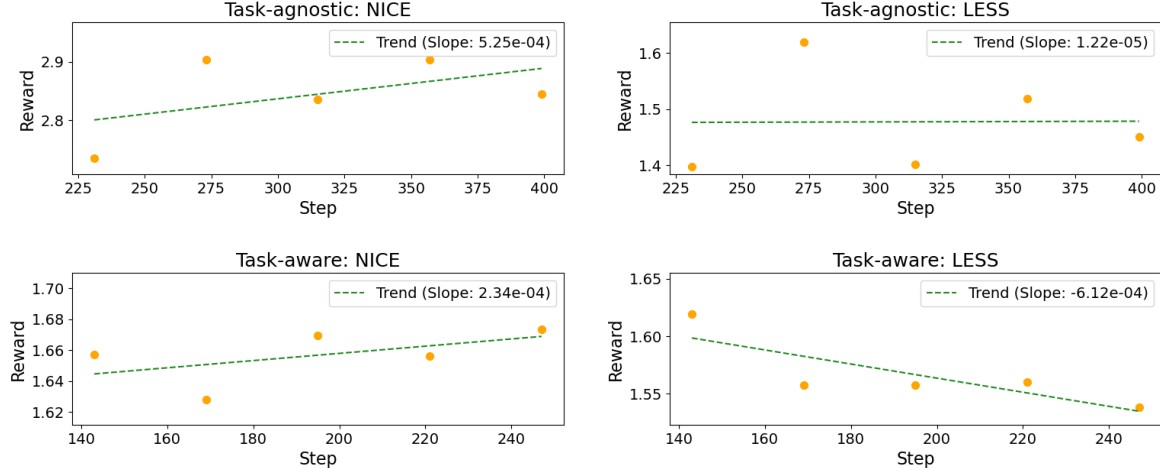

*Figure 2.* Validation performance, measured by the reward model of the last few checkpoints, for models trained on subsets selected by NICE and LESS for the RLHF task. NICE-selected data optimizes for improved validation performance, whereas LESS-selected data focuses on loss reduction, which may not always enhance validation performance.

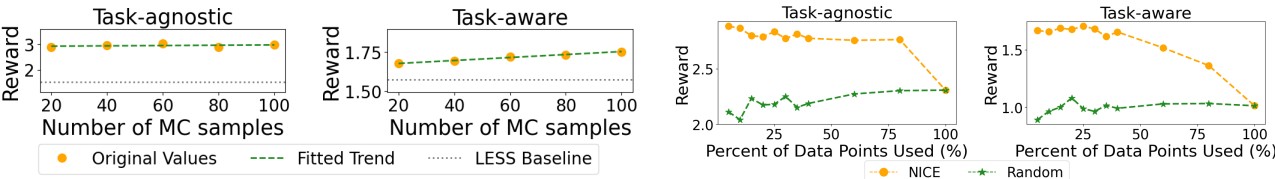

*Figure 3.* Performance of LLMs trained on data selected by NICE for RHLF task when different numbers of MC samples are used. There is a positive correlation between performance and generated MC samples. Using the sampling size of 20 provides good performance while increasing the sampling size has the potential to improve the performance.

*Figure 4.* Performance versus percent of data points selected by NICE or Random for the RLHF task. As the percentage selected by NICE increases, performance may drop due to the inclusion of low-score, irrelevant, or harmful points. When using data selection, the trained model consistently outperforms those trained on randomly selected data, even outperforming the full dataset, underscoring the importance of data selection.

tions tasks whose evaluation metrics align poorly with NTP loss, as elaborated in Sec. 4.5. On the other hand, the works of Kwon et al. (2024); Choe et al. (2024); Lin et al. (2024) focus on data attribution and hence do not optimize for data selection. In addition, Park et al. (2023) approximates the data model to estimate the influence; Wang et al. (2024a) scales up the Shapley value to estimate the influence; and Qiao et al. (2025) uses the influence function to reweight the training samples. However, these methods only focus on classification tasks. In contrast, our work focuses on the influence estimation for tasks that require extensive generations. More related works on data curation for instruction tuning are discussed in App. G.

than relying on NTP loss. When using a reward function that does not require label response, NICE can perform data selection without relying on costly annotated labels. Experimental results show that our approach outperforms existing data selection methods across diverse scenarios. Of note, despite the superior performance achieved by NICE and NICE$_{\text{AMC}}$, the computational cost of these approaches is not negligible even with the acceleration (e.g., the use of LoRA and random projection). Further explorations can be done to study other computationally efficient ways of computing the gradients.

## 6. Conclusion

We propose NICE, a novel influence estimation approach that selects training data to directly optimize non-differentiable evaluation metrics via policy gradient, rather

## Impact Statement

This paper presents work whose goal is to advance the field of Machine Learning. There are many potential societal consequences of our work, none of which we feel must be specifically highlighted here.

## Acknowledgments

This research/project is supported by the National Research Foundation, Singapore under its National Large Language Models Funding Initiative (AISG Award No: AISG-NMLP-2024-001). Any opinions, findings and conclusions or recommendations expressed in this material are those of the author(s) and do not reflect the views of National Research Foundation, Singapore. This research is supported by the National Research Foundation Singapore and the Singapore Ministry of Digital Development and Innovation, National AI Group under the AI Visiting Professorship Programme (award number AIVP-2024-001). This research is supported by the National Research Foundation (NRF), Prime Minister's Office, Singapore, under its Campus for Research Excellence and Technological Enterprise (CREATE) programme. The Mens, Manus, and Machina (M3S) is an interdisciplinary research group (IRG) of the Singapore MIT Alliance for Research and Technology (SMART) centre. Jingtan Wang is supported by the Institute for Infocomm Research of Agency for Science, Technology and Research (A*STAR). We would like to thank the anonymous reviewers and AC for the constructive and helpful feedback.

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

# A. Training

## A.1. Training Dataset

For the task-agnostic setting, the four processed training datasets that are utilized are described in Wang et al. (2023). These datasets, annotated or authored by humans, are detailed in Tab. 7. FLAN V2 and COT are based on existing NLP datasets, while DOLLY and OASST feature open-ended responses from humans, demonstrating diverse formats, lengths, and tasks.

For the task-aware setting, we use 95% of the *RLHF* task's training data as our training set, while the remaining 5% serves as the validation set. The RLHF dataset is constructed from conversations between humans and AI assistants. Specifically, given an existing (potentially multi-turn) conversation between an LLM and a human, the LLM provides two responses, which are then evaluated by human annotators. Each data point contains a pair of responses, with one marked as 'chosen' by human annotators for being more helpful than the other. Since only the last responses from the assistant are compared during human labeling, the helpfulness is evaluated only for the last-turn response of each conversation. Consequently, we use only the last-turn response as labels. We only use these "chosen" responses for training, validation, and test. For the *HumanEval* task's training data, we adopt the CodeAlpaca 20k (Chaudhary, 2023) dataset with the addition of the original instruction format for each data point.

For fine-tuning Llama2, we adopt the 'Tulu' format following from the study by Wang et al. (2023).

---

**Tulu Instruction Format**

```
<user>
```
What can you help me with?

```
<assistant>
```
I'd like to show off how chat templating works!

---

For fine-tuning the Mistral and Llama3 models, we utilize the respective instruction formats predefined by each model, as detailed below:

---

**Mistral Instruction Format**

`[INST]` What can you help me with? `[/INST]` I'd like to show off how chat templating works!

---

**Llama3 Instruction Format**

`<|start_header_id|>user<|end_header_id|>`

What can you help me with?`<|eot_id|><|start_header_id|>assistant <|end_header_id|>`

I'd like to show off how chat templating works!

---

## A.2. Training Details

All experiments utilized the parameter-efficient LoRA (Hu et al., 2021) approach. A linear warm-up with $0.03$ warmup ratio was employed, peaking at a learning rate of $2 \times 10^{-5}$. We trained for $4$ epochs on each dataset with a batch size of 128. The LoRA module had a rank of 128, an $\alpha$ of 512, a dropout rate of 0.1, and learned matrices for all attention layers. Specifically, the Llama2-7b model used is `meta-llama/Llama-2-7b-hf`, Llama2-13b used is `meta-llama/Llama-2-13b-hf`, Mistral-7B is `mistralai/Mistral-7B-v0.1`, Llama3-8B is `meta-llama/Meta-Llama-3-8B`.

Each main experiment was repeated three times with random seeds 0, 1, and 2. Under random selection methods, three different random subsets from the training set were chosen for each seed. For LESS, TSDS, NICE, and NICE_AMC_, we first performed warmup training on subsets chosen by each seed, then selected distinct subsets from the resulting warmup model for each trial. We used the same optimization seeds as those used for the warmup model. For experiments without reported standard deviation, we used seed 0. Full training was conducted on seed 0 only, due to heavy computation.

*Table 7.* Detailed information about the training set. The task-agnostic training data pool is the same as in Xia et al. (2024) and Wang et al. (2023).

| Task-agnostic | | | | | |
|---|---|---|---|---|---|
| **Dataset** | **Size** | **Sourced from** | **# Turns** | **Prompt Len.** | **Response Len.** |
| FLAN V2 | 100,000 | Based on Existing NLP Datasets | 1 | 355.7 | 31.2 |
| COT | 100,000 | Based on Existing NLP Datasets | 1 | 266.0 | 53.2 |
| DOLLY | 15,011 | Human-written from scratch | 1 | 118.1 | 91.3 |
| OASST | 55,668 | Human-written from scratch | 1.6 | 34.8 | 212.5 |
| Task-aware | | | | | |
| RLHF | 41,643 | Human feedback data | 2.46 | 145.0 | 517.3 |
| Code Alpaca | 20,022 | Code-related human-written instructions | 1 | 294.8 | 197.0 |

# B. Evaluation

## B.1. Evaluation Dataset

**AlpacaEval** sources its data from self-instruct, OASST, Anthropic's helpful dataset, Vicuna, and Koala, widely used for understanding model behavior in structured, instruction-driven settings. The evaluation metric is a length-controlled win rate, adjusting for biases in response length using a regression model to ensure fair and accurate assessments (Dubois et al., 2024). Each response is compared to a baseline model, *text_davinci_003*, using *weighted_alpaca_eval_gpt4_turbo* as the annotator. We use *text_davinci_003* as a baseline because its relatively lower baseline performance can more clearly highlight the performance difference between selected subsets. By contrast, using a stronger baseline (e.g., *gpt4_turbo*) could mask the differences among the model trained on the subset selected by different selection strategies. For annotation, we employ *weighted_alpaca_eval_gpt4_turbo* (https://github.com/tatsu-lab/alpaca_eval), chosen for its high agreement with human annotations, large context capacity, and cost-effectiveness.

**HumanEval** uses a set of programming challenges to evaluate code correctness and functionality, measured by the pass@$k$ metric which is how many correct solutions appear within a specified number of attempts $k$ (e.g., we use 100 in the main experiment). This metric assesses the ability of models to generate correct solutions within a limited number of attempts, reflecting the model's efficiency in code generation. We additionally use the unbiased estimator of pass@$k$ to avoid the high variance of vanilla pass@$k$ (Chen et al., 2021).

**TLDR** uses cleaned data from Stiennon et al. (2020), focusing on well-structured input-output pairs for summarization tasks. It is evaluated against a reward model (OpenAssistant, 2023) that is trained based on human feedback to ensure the generation of high-quality summaries.

**RLHF** is designed for training and evaluating language models using human feedback to optimize response generation. It consists of prompt-response pairs and we only use the "chosen" response whose last-turn response aligns best with human preferences, according to specific criteria such as relevance and safety. The evaluation metric is reward model (Ray2333, 2024) which can measure the helpfulness of model responses.

## B.2. Evaluation Details

For AlpacaEval and HumanEval, we set the number of tasks to be $q = 5$ and $q = 10$, while for TLDR and RLHF have $q = 1$. The reason is: for AlpacaEval, two samples are drawn from each of the five subtasks (Self-Instruct, OASST, Anthropic's helpful dataset, Vicuna, Koala), resulting in a total of ten samples in the validation set. For HumanEval, since pass@$k$ aims to check if there is at least one functional code within the $k$ generations of each test data point, we select training data points that enhance performance across all validation data. Each validation data point is treated as an individual task. Hence, AlpacaEval and HumanEval are multi-subtask scenarios with $q = 5$ and $q = 10$, while TLDR and RLHF are single-subtask scenarios with $q = 1$. The aggregation of each training data point's influence score w.r.t the validation set follows from Sec. 2.2.

For TLDR and RLHF, the reward function $r$ used during the computation of the policy gradient is the reward model, and the

evaluation metric for the validation set is the average reward for each validation point. For AlpacaEval, the reward function $r$ for each generated response is the annotator's decision for that response, and the evaluation metric for the validation set is not simply an average but an average of a debiased version of each annotator's decision (Dubois et al., 2024). For HumanEval, the reward function $r$ for each generated response (code) is the boolean result of all unit tests for that code, while the evaluation metric for the validation set is pass@$k$. This metric measures functional correctness by generating $k$ codes for each test problem and considering the problem solved if any code passes all unit tests of that problem. The pass@1 score can be viewed as the average of each boolean result, whereas pass@10 and pass@100 are computed via a problem-level "OR" across the $k$ generated codes, followed by an average across all problems. Additionally, to address numerical instability and reduce variance, we use the unbiased estimator version of pass@$k$ following Chen et al. (2021).

We evaluate the model on the validation set $D_V$ (the same set used for data selection) after each epoch, and assess the best-performing checkpoint on the test set.

## C. NICE and NICEAMC Details

NICE and NICEAMC rely solely on the probability of generated responses and the score from a reward function, as enabled by the policy gradient mechanism. For AlpacaEval, labels are provided to all approaches because the reward function requires ground-truth labels. For the other three tasks, including TLDR, RLHF, and HumanEval, the ground-truth labels of the validation dataset are not used by our method, although they are available to the other baselines (such as LESS). This is because in these three tasks, the reward function does not require ground-truth labels. Specifically:

- For RLHF and TLDR, the reward function is a learned reward model that outputs scores based on the prompt and generated response.

- For HumanEval, the reward is defined by whether the generated code passes unit tests, not requiring reference solutions.

### C.1. NICE Details

We generate 20 MC samples for all evaluation datasets, except for HumanEval where we generate 500 samples due to the task's difficulty (i.e., a lower code pass rate). Generally, we set the sampling temperature to 1.2 to promote diversity, except for AlpacaEval where we use 1.0. The generated responses of the validation set under temperature of 1.0 yield a higher win rate on the final checkpoint compared to 1.2. We employ multinomial sampling with `top_k` = 50 and `top_p` = 0.95.

### C.2. NICEAMC Details

For NICEAMC, we use `gpt-4-turbo-2024-04-09`. Regarding the GPT API hyperparameters, we set the frequency_penalty to 0, presence_penalty to 0, and temperature to 0.8 for all tasks.

For TLDR tasks, we add a prompt `A brief summary of my post is (TL;DR):` after the prompt of the data point, before generating the response, to enhance generation quality.

For HumanEval, we prepend a prompt:

```
Complete the following python function to return only the function body (completion).
Do not include the function header or docstring.
```

before the coding question. This ensures the model outputs only the necessary code, avoiding chain-of-thought content that could fail unit tests.

## D. Baseline Details

For LESS and TSDS, the hyperparameter settings are the same as in their official repos. For BM25, DSIR, and RDS, these methods are warmup model-agnostic, meaning the selection process does not rely on the warmup models' randomness. Hence, we only run them on seed 0. To avoid the instruction format's effect on the representation or retrieval, we use the format of:

```
Question:[Question]\n\nAnswer:[Answer]
```

for the majority of the training data and validation data. We use TULU format for RLHF training and validation data, as RLHF contains many turns.

For BM25, we use the rank_bm25 package (https://github.com/dorianbrown/rank_bm25). We treat each validation data point as a query to retrieve the BM25 scores of each training data point. For RDS, we adopt the sentence-transformers/all-MiniLM-L6-v2 model (https://huggingface.co/sentence-transformers/all-MiniLM-L6-v2). We compute the cosine similarity of embeddings between each validation data point and each training data point. Overall, for each validation point, BM25 and RDS will have a score vector of dimension $n$ (the size of the training dataset is $n$). We then follow the same aggregation steps as in Sec. 2.2 to aggregate the scores. For DSIR, we use the official Github repo and match the distribution between training data and validation data (https://github.com/p-lambda/dsir).

# E. Preliminary

### E.1. Derivation of The Change in the Validation Loss

The detailed derivation of the change in the validation loss is explained here: When a training data point $z_i$ is included in the training step $t$, the model parameters are updated accordingly, leading to a change in the validation loss. Assuming a small learning rate $\eta_t$ is used in the parameter updates with the Stochastic Gradient Descent (SGD) optimizer, this one-step change at step $t$ can be approximated using a first-order Taylor expansion (Pruthi et al., 2020):

$$
\begin{aligned}
&L(z_v; \theta^{t+1}) - L(z_v; \theta^t) \\
&= \nabla_\theta L(z_v; \theta^t) \cdot (\theta^{t+1} - \theta^t) + O(\|\theta^{t+1} - \theta^t\|^2) \\
&\approx \nabla_\theta L(z_v; \theta^t) \cdot (\theta^{t+1} - \theta^t) \\
&= \nabla_\theta L(z_v; \theta^t) \cdot (-\eta_t \nabla_\theta L(z_i; \theta^t)) \\
&= -\eta_t \langle \nabla_\theta L(z_v; \theta^t), \nabla_\theta L(z_i; \theta^t) \rangle .
\end{aligned}
\tag{1}
$$

### E.2. Derivation of Influence Function

Influence Function (IF) (Koh & Liang, 2017) measures the influence of down-weighting the training data point $z_i$ by some small $\epsilon$, on the new parameter $\theta^E_{\epsilon,z_i} := \arg\min_{\theta \in \Theta} \frac{1}{n} \sum_{i=1}^n L(z_i; \theta) + \epsilon L(z_i; \theta)$. The parameter change is given by:

$$
\mathcal{I}_{\text{down,params}}(z_i) = \left. \frac{\partial \theta^E_{\epsilon,z_i}}{\partial \epsilon} \right|_{\epsilon=0} = H_{\theta^E}^{-1} \nabla_\theta L(z_i; \theta^E)
$$

where $H_{\theta^E} = \frac{1}{n} \sum_{i=1}^n \nabla_\theta^2 L(z_i; \theta^E)$ is the Hessian matrix of the average loss over the training set and is positive definite by assumption. Then, by applying the chain rule, IF can measure the influence of down-weighting $z_i$ on the loss of the validation data point $z_v$:

$$
\begin{aligned}
\text{Inf}_{\text{IF}}(z_i, z_v) &= \left. \frac{\partial L(z_v; \theta^E_{\epsilon,z_i})}{\partial \epsilon} \right|_{\epsilon=0} \\
&= \nabla_\theta L(z_v; \theta^E)^\top \left. \frac{\partial \theta^E_{\epsilon,z_i}}{\partial \epsilon} \right|_{\epsilon=0} \\
&= \nabla_\theta L(z_v; \theta^E)^\top H_{\theta^E}^{-1} \nabla_\theta L(z_i; \theta^E) .
\end{aligned}
\tag{2}
$$

### E.3. Efficient Influence Function

DataInf makes use of Bartlett's second identity (Bartlett, 1953) to approximate the Hessian. That is, the Hessian can be replaced with the second moment of the first-order gradients: $G(\theta^E) = \frac{1}{n} \sum_{i=1}^n \nabla_\theta L(z_i; \theta^E) \nabla_\theta L(z_i; \theta^E)^\top$, which simplifies the computation of the Hessian matrix. The $\theta^E$ here is the last checkpoint of the warmup model. In the later section, the computation of DataInf is all w.r.t. to the last checkpoint of the warmup model. We denote the gradient of the loss of training data point $z_i$ w.r.t. $\phi$ ($\phi$ can be a single layer's parameter) by $\nabla_\phi L_i := \nabla_\phi L(z_i; \theta^E)$, while the gradient of the loss of validation data point $z_v$ w.r.t. $\phi$ ($\phi$ can be a single layer's parameter) by $\nabla_\phi L_v := \nabla_\phi L(z_v \theta^E)$.

To further address the potential large dimension of $\theta$ and computational challenges, DataInf adopts another two techniques: 1. **Damping**: A small positive constant, $\lambda$, is added to the diagonal elements of $G(\theta^E)$, enhancing its positive definiteness

and invertibility (Martens et al., 2010). 2. **Block Diagonal Matrix Representation**: $G(\theta^E)$ is further approximated using its block diagonal matrix where each block is a layer of the deep neural network (Grosse et al., 2023). The influence function then becomes

$$\sum_{l=1}^{L} \nabla_{\theta_l} L_v^{\top} \left( \frac{1}{n} \sum_{j=1}^{n} \nabla_{\theta_l} L_j \nabla_{\theta_l} L_j^{\top} + \lambda I_{d_l} \right)^{-1} \nabla_{\theta_l} L_i$$

where $\theta_l \in \mathbb{R}^{d_l}$ is the model parameter in $l$-the layer, $I_{d_l} \in \mathbb{R}^{d_l \times d_l}$ is identify matrix of size $d_l$, and $L$ is the total number of layers.

Following these transformations, DataInf tweaks the order of matrix inversion and the average calculations, and then leverages the Sherman-Morrison formula:

$$\left( \frac{1}{n} \sum_{j=1}^{n} \nabla_{\theta_l} L_j \nabla_{\theta_l} L_j^{\top} + \lambda I_{d_l} \right)^{-1} \approx \frac{1}{n} \sum_{j=1}^{n} \left( \nabla_{\theta_l} L_j \nabla_{\theta_l} L_j^{\top} + \lambda I_{d_l} \right)^{-1}$$

$$= \frac{1}{n\lambda} \sum_{j=1}^{n} \left( I_{d_l} - \frac{\nabla_{\theta_l} L_j \nabla_{\theta_l} L_j^{\top}}{\lambda + \nabla_{\theta_l} L_j^{\top} \nabla_{\theta_l} L_j} \right) .$$

The computation of the Hessian and the later computation of the influence function can require retrieving the training gradient $\nabla_\phi L_i$ twice. To additionally optimize the time, we apply random projections to each gradient vector $\nabla_\phi L_i$ and store them. Later, we can retrieve the stored projected gradients and compute Hessian and the influence function score.

Besides *DataInf*, other methods have been proposed for efficiently computing the influence function for large language models. Grosse et al. (2023) improved the computation of the block-diagonal Hessian using the Kronecker product of uncentered forward and backward covariances of each layer (*EK-FAC*). *TRAK* projects gradients into a low-dimensional space and calculates influence scores within the subspace (Park et al., 2023). *LORGA* further improves the projection step with an efficient gradient projection strategy that leverages the gradient structure in backpropagation (Choe et al., 2024). However, neither *EK-FAC* nor *LORGA* discussed their applicability to *LoRA* fine-tuned models, leading us to exclude them for experiment. Additionally, *TRAK*'s approach of treating the multi-class classification problem as a single binary logistic regression may result in information loss. Therefore, we also did not apply it, as our task is purely generative and each token's prediction is a multi-class classification prediction whose prediction space is vocabulary size. For the hyperparameter in terms of Datainf, the projected gradient is 2048 dimension. The smaller dimension is due to the fact that we need to project gradients for each layer. The $\lambda$ is 0.0001.

## F. Explicit form of NICE During Implementation

As discussed in Sec. 3.1, Xia et al. (2024) replaces the SGD with Adam gradient and replaces the inner product with the cosine similarity of the original TracIn for performance consideration. We integrate these two enhancements on NICE as well, leading to:

$$\text{Inf}_{\text{NICE}}(z_i, z_v) = \sum_{e=1}^{E} \bar{\eta}_e \frac{\left\langle \mathbb{E}_{\hat{y}_v \sim f(y|x_v; \theta^e)} \left[ -\nabla_\theta \log(f(\hat{y}_v|x_v; \theta^e)) r(z_v, \hat{y}_v) \right], \Gamma_\theta(z_i; \theta^e) \right\rangle}{\left\| \mathbb{E}_{\hat{y}_v \sim f(y|x_v; \theta^e)} \left[ -\nabla_\theta \log(f(\hat{y}_v|x_v; \theta^e)) r(z_v, \hat{y}_v) \right] \right\| \left\| \Gamma_\theta(z_i; \theta^e) \right\|}$$

$$\Gamma_\theta(z_i; \theta^e) \triangleq \frac{\mathbf{m}^{e+1}}{\sqrt{\mathbf{v}^{e+1} + \epsilon}}$$

$$\mathbf{m}^{e+1} = \frac{\beta_1 \mathbf{m}^e + (1 - \beta_1) \nabla_\theta L(z_i; \theta^e)}{1 - \beta_1^e}$$

$$\mathbf{v}^{e+1} = \frac{\beta_2 \mathbf{v}^e + (1 - \beta_2) \left( \nabla_\theta L(z_i; \theta^e) \right)^2}{1 - \beta_2^e}$$

where every operation is applied elementwise. Here, $\beta_1$ and $\beta_2$ represent the hyperparameters for the first and second moments, respectively, with $\epsilon$ serving as a small constant.

# G. Related Work: Data curation for instruction tuning

Curating high-quality, diverse, and complex instruction tuning data has been shown to improve the instruction-following ability of LLMs. Researchers have proposed different strategies to measure and improve different aspects of instruction-following ability: Cao et al. (2023) utilize natural language indicators to evaluate quality; Zhao et al. (2024) employ GPT-transformed instructions to measure complexity; and Chen et al. (2023) adopt LLM-annotated scores to assess both quality and complexity. Bukharin & Zhao (2023); Du et al. (2023); Lu et al. (2024); Wang et al. (2024b) optimize instruction data by emphasizing diversity in tandem with quality or complexity. Liu et al. (2024a) offers a comprehensive comparison of existing methods regarding these three properties. They further train a model to predict complexity and quality and iteratively filter out the most diverse points using embeddings. These approaches typically select data without a validation set. Our work extends this line of research to targeted data selection with guidance from validation set, selecting data that aligns the most with downstream tasks to enhance specific model capabilities. Our setting is more similar to Xia et al. (2024), Liu et al. (2024b) and Li et al. (2024b). However, Xia et al. (2024) selects data based on the influence on validation loss, which can lead to discrepancies between minimizing loss and maximizing performance, while we select data based on their influence on validation performance. Liu et al. (2024b) uses loss-based gradients to measure data distance when optimizing for distribution alignment and diversity, which can also suffer from the aforementioned discrepancy to some extent. Li et al. (2024b) utilizes a perplexity-based scoring system to select the most advantageous data for a defined anchor set, but their methodology is limited to single-turn training data. Another line of work, exemplified by Bhatt et al. (2024), frames curation as active learning by selecting the most informative prompts for predicting the label (i.e., generate responses) through uncertainty or diversity maximization. This active learning paradigm differs from our setting. A more comprehensive review of data curation methods is provided by Albalak et al. (2024).

# H. Additional Analysis

Unless otherwise specified, the experiments in the section below are conducted on the Llama2-7B model.

## H.1. Additional Results of Pass@$k$ on HumanEval

We provide additional evaluation metrics for HumanEval dataset, specifically pass@1 and pass@10, in Tab. 8. Overall, they align with our main findings: NICE and NICEAMC generally outperform loss-based influence estimation (LESS) and baselines, and the subsets selected by NICE and NICEAMC have the potential to outperform the full dataset. Note that BM25 performs well in terms of pass@1 and pass@10 on the Mistral-7B model and even outperforms other baselines under a task-agnostic training data pool. Empirically, we observe that BM25 is more likely to select data points that contain codes. This is likely because BM25 is based on TF-IDF, which assigns higher scores to training data that is more relevant in terms of word frequency, and certain words appear more frequently than others in the codes. Consequently, BM25 performs well, especially when the training data contains data from multiple different domains that are not coding-related. However, this good result does not transfer to either the Llama2-7B model or under a task-aware training data pool due to two main reasons: 1) BM25 does not use the information from the models, meaning that the same data subset will be selected for different models. Intuitively, different models require different data to achieve better performance. Therefore, selecting data using BM25 is sub-optimal; 2) Selection based on the word frequency is not enough for the task-aware setting. In a task-aware data pool, training data points are more relevant to the task, possibly resulting in comparable BM25 scores for all data points. Consequently, a more careful selection based on other criteria (i.e., not just word frequency) is needed. Additionally, while BM25 achieves a 31.81 pass@1 for Mistral-7B, pass@1 is not the sole evaluation criterion in practice, because there are situations where multiple responses can be generated from the LLM and checked by a verifier (e.g., test cases). Therefore, for HumanEval, focusing on pass@$k$ with larger $k$ (and higher accuracy) is more desirable.

Additionally, we argue that NICE can improve pass@1 performance with lower temperature while sacrificing the performance of pass@10 and pass@100. We analyze the performance on the HumanEval task for models trained on NICE-selected subsets when using different temperatures to generate MC samples. As shown in Tab. 9, the pass@1 performance on downstream tasks is improved by using a lower temperature, while at the cost of reduced performance on pass@10 and pass@100 metrics. A lower temperature reduces uncertainty during generation. If the model is good at certain problems, it increases the probability of answering these problems correctly. However, this reduction in uncertainty comes at the expense of diversity, as the generated responses tend to be very similar to one another. Consequently, for difficult questions, if all generated responses are incorrect, pass@$k$ (for larger k) suffers. This trade-off implies that using a lower temperature to generate responses improves pass@1 performance. When these responses generated under a lower temperature are used to

*Table 8.* Additional Results of Pass@k on HumanEval for both *task-agnostic* and *task-aware* settings on Llama2-7B and Mistral-7B. **Bold** numbers indicate the top-performing selected subset. A purple cell suggests that NICE outperforms LESS which uses loss-based influence estimation. Underlined numbers show that the subset selected by our approach exceeds the performance of the full dataset. Numbers in small font represent standard deviations.

**Task-agnostic training data pool.**

|  | Llama2-7B | | | Mistral-7B | | |
|---|---|---|---|---|---|---|
|  | Pass@1 | Pass@10 | Pass@100 | Pass@1 | Pass@10 | Pass@100 |
| Full | 7.61 | 25.52 | 47.44 | 29.47 | 59.68 | 83.63 |
| Random | $8.34_{\pm0.34}$ | $23.85_{\pm0.40}$ | $44.30_{\pm2.36}$ | $29.99_{\pm1.37}$ | $62.04_{\pm1.57}$ | $85.56_{\pm1.27}$ |
| RDS | 10.00 | 25.55 | 45.29 | 30.31 | 62.00 | 84.15 |
| BM25 | 8.27 | 24.51 | 46.19 | **31.81** | **62.43** | 84.09 |
| DSIR | 9.53 | 24.02 | 42.22 | 27.71 | 56.81 | 79.17 |
| TSDS | $10.30_{\pm1.58}$ | $25.47_{\pm1.27}$ | $43.68_{\pm1.82}$ | $27.50_{\pm1.47}$ | $59.78_{\pm1.86}$ | $82.78_{\pm1.25}$ |
| LESS | $9.24_{\pm0.77}$ | $26.12_{\pm0.17}$ | $47.50_{\pm1.57}$ | $26.85_{\pm0.58}$ | $60.66_{\pm0.39}$ | $85.24_{\pm0.45}$ |
| NICE | $\underline{\mathbf{10.35}}_{\pm1.72}$ | $\underline{\mathbf{27.37}}_{\pm1.56}$ | $\underline{\mathbf{48.59}}_{\pm2.08}$ | $\underline{29.48}_{\pm0.93}$ | $\underline{62.05}_{\pm2.23}$ | $\underline{85.59}_{\pm1.41}$ |
| NICE_AMC | $\underline{9.04}_{\pm2.35}$ | $25.11_{\pm1.72}$ | $45.10_{\pm2.84}$ | $\underline{29.96}_{\pm1.95}$ | $\underline{62.10}_{\pm1.82}$ | $\underline{\mathbf{85.67}}_{\pm0.34}$ |

**Task-aware training data pool.**

|  | Llama2-7B | | | Mistral-7B | | |
|---|---|---|---|---|---|---|
|  | Pass@1 | Pass@10 | Pass@100 | Pass@1 | Pass@10 | Pass@100 |
| Full | 13.27 | 30.30 | 51.27 | 33.14 | 64.09 | 84.27 |
| Random | $11.99_{\pm0.22}$ | $29.86_{\pm0.42}$ | $51.91_{\pm1.61}$ | $33.15_{\pm0.76}$ | $63.62_{\pm1.92}$ | $83.34_{\pm2.54}$ |
| RDS | 12.40 | 31.34 | 54.74 | 33.32 | 63.23 | 86.75 |
| BM25 | **13.66** | 31.22 | 52.23 | 33.58 | 64.35 | 84.81 |
| DSIR | 11.98 | 30.43 | 53.10 | 32.30 | 59.07 | 79.91 |
| TSDS | $12.85_{\pm0.50}$ | $28.15_{\pm0.91}$ | $49.85_{\pm3.17}$ | $31.74_{\pm1.63}$ | $63.04_{\pm1.06}$ | $85.51_{\pm1.28}$ |
| LESS | $13.55_{\pm0.28}$ | $30.53_{\pm0.57}$ | $52.67_{\pm0.71}$ | $34.05_{\pm1.28}$ | $64.12_{\pm0.37}$ | $85.26_{\pm1.13}$ |
| NICE | $\underline{13.43}_{\pm0.33}$ | $\underline{\mathbf{31.70}}_{\pm0.66}$ | $\underline{\mathbf{55.09}}_{\pm1.66}$ | $\underline{33.61}_{\pm1.29}$ | $\underline{\mathbf{65.56}}_{\pm1.32}$ | $\underline{\mathbf{87.35}}_{\pm1.03}$ |
| NICE_AMC | $12.87_{\pm0.53}$ | $\underline{30.39}_{\pm0.26}$ | $50.67_{\pm1.24}$ | $\underline{\mathbf{34.13}}_{\pm0.88}$ | $63.91_{\pm0.84}$ | $84.18_{\pm1.63}$ |

*Table 9.* Performance on the HumanEval task under a task-aware training data pool for models trained on NICE-selected subsets with different temperatures for generating MC samples. We can decrease the temperature to improve pass@1 metric, while at the cost of decreased performance on pass@10 and pass@100.

| Model | Temperature 1.0 | | | Temperature 1.2 | | |
|---|---|---|---|---|---|---|
|  | pass@1 | pass@10 | pass@100 | pass@1 | pass@10 | pass@100 |
| Llama2-7B | 14.13 | 31.21 | 49.95 | 13.12 | 31.41 | 53.96 |
| Mistral-7B | 35.70 | 63.60 | 81.04 | 35.23 | 63.85 | 85.69 |

compute policy gradients for data selection, the resulting selected subset also favors the pass@1 metric, while at a cost of decreased performance on pass@$k$ (for larger k).

### H.2. Robustness across Validation Splits

We demonstrate the robustness of NICE across different validation splits and address concerns about potential overfitting to a specific validation set. When the training data pool is task-agnostic, we randomly selected an alternative validation set $D'_V$ as a reference and re-selected a subset $D'_S$. The performance of models retrained on $D_S$ (selected based on the original validation split $D_V$) and $D'_S$ (selected based on the new validation split $D'_V$) are shown in Fig. 5. Importantly, models trained on the selected subsets, whether $D_S$ or $D'_S$, consistently outperform models trained on randomly selected subsets. This verifies the robustness of our approach and confirms that its effectiveness does not depend on a specific validation split.

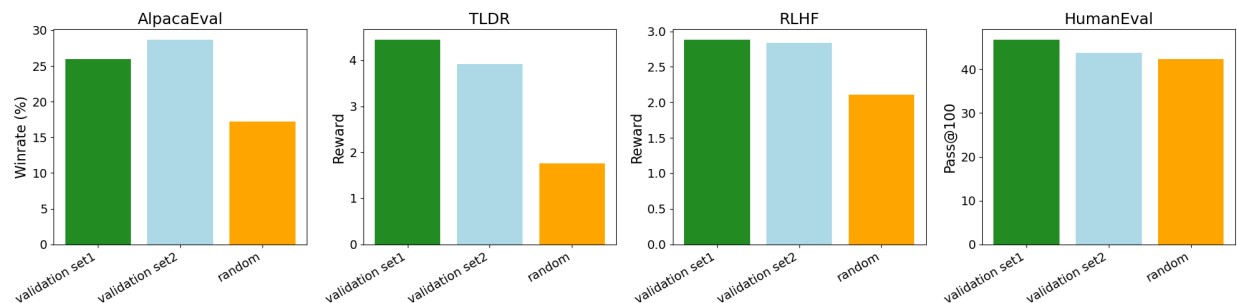

*Figure 5.* Performance of models trained on different NICE-selected subsets using different validation sets as references. Models trained on NICE-selected subsets consistently outperform those trained on randomly selected subsets, regardless of the validation set used.

*Table 10.* Additional Results on the RLHF task for Llama2-13B and Llama3-8B. **Bold** numbers indicate the top-performing selected subset. A purple cell suggests that NICE outperforms LESS which uses loss-based influence estimation.

| RLHF | Llama2-13B | | Llama3-8B | |
|---|---|---|---|---|
| | Task-agnostic | Task-aware | Task-agnostic | Task-aware |
| Random | $2.06_{\pm 0.04}$ | $1.20_{\pm 0.07}$ | $1.97_{\pm 0.07}$ | $1.12_{\pm 0.06}$ |
| RDS | 1.77 | 0.70 | 1.75 | 0.81 |
| BM25 | 2.72 | 1.34 | 2.84 | 1.43 |
| LESS | $1.52_{\pm 0.09}$ | $1.65_{\pm 0.04}$ | $1.64_{\pm 0.14}$ | $1.65_{\pm 0.08}$ |
| NICE | $\mathbf{2.87}_{\pm 0.04}$ | $\mathbf{1.76}_{\pm 0.04}$ | $\mathbf{3.22}_{\pm 0.02}$ | $\mathbf{1.99}_{\pm 0.06}$ |

## H.3. Additional results on Llama3-8B and Llama2-13B

We evaluate NICE against various data selection baselines using the state-of-the-art model, Llama3-8B (Dubey et al., 2024), and a larger model, Llama2-13B, on the RLHF task (see Tab. 10). The superiority of NICE underscores our method's generalizability across different model sizes and state-of-the-art models.

## H.4. Discrepancy between NTP Loss and Evaluation Metrics

We additionally include the NTP loss and performance (i.e., measured by each task's evaluation metric here) of the validation set for the remaining three tasks in several training checkpoints. The results in Fig. 6 are similar to those in Fig. 1: checkpoints with minimal loss (highest negative losses) do not correspond to checkpoints with the best performance; the performance can continue to increase even if the loss increases (negative loss decreases).

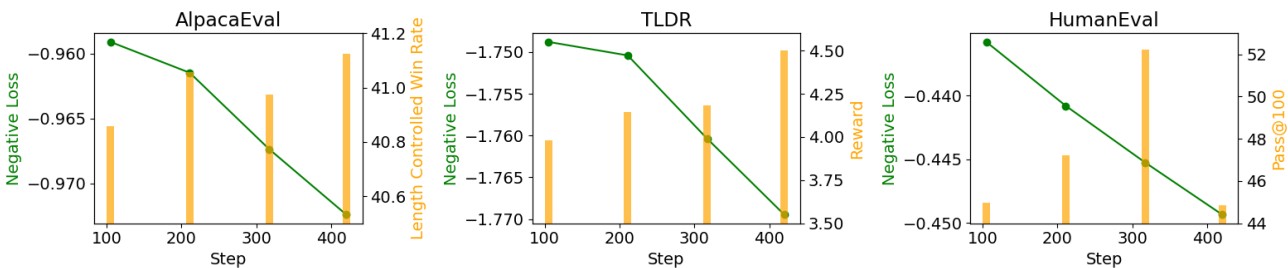

*Figure 6.* Discrepancy observed between the NTP loss and performance (as measured by each task's evaluation metric) on the checkpoints. The checkpoint with the lowest loss (i.e., most negative) can exhibit relatively poorer performance. Notably, performance can continue to improve even as the loss worsens (i.e., the negative loss becomes higher).

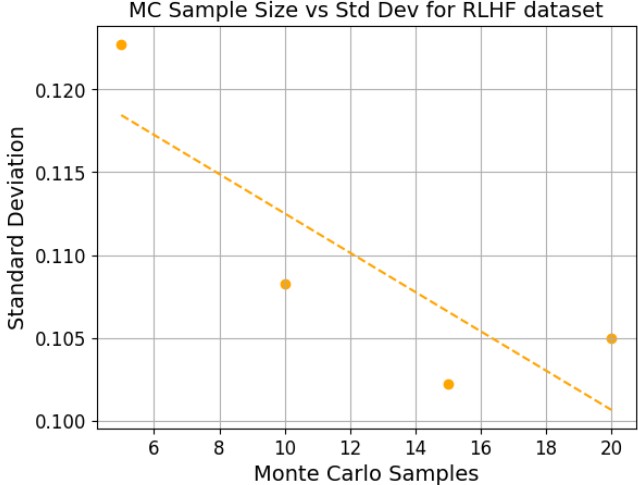

*Figure 7.* Standard deviation of different runs w.r.t the number of MC samples. Increasing the number of MC samples generally lowers the standard deviation across runs, indicating better stability.

### H.5. Stability of MC Sampling

We provide an ablation study (Fig. 7), varying MC samples from 5 to 20 on the RLHF task under the task-agnostic training data pool, for a more in-depth discussion on stability. Results show that increasing the number of MC samples generally lowers the standard deviation across runs with different seeds, indicating better stability. The benefit of reduced standard deviation diminishes as it increases. This validates that our chosen MC (MC=20 for RLHF task) provides a good trade-off, offering sufficient stability without excessive computation.

### H.6. On the Additional Cost of NICE_AMC

Note that NICE_AMC is an optional enhancement—NICE itself does not require GPT-4. We list the projected GPT-4 cost for NICE_AMC in Tab. 11. The costs are low for the majority of the tasks, except for RLHF, due to its large validation set (which can be addressed by using alternative models as discussed in the next paragraph).

*Table 11.* Projected GPT-4 cost for NICE_AMC across different tasks. The cost for RLHF is high due to a large validation set.

| Task | AlpacaEval | TLDR | RLHF | HumanEval | Avg |
|---|---|---|---|---|---|
| GPT Cost ($) | 1.70 | 14.26 | 291.17 | 6.34 | 78.37 |

**Use of Open-Source/Smaller LLMs.** To reduce cost, we can use high-performing open-source models. On the RLHF task in the task-agnostic setting, we use Qwen 2.5-3B/7B-Instruct (Yang et al., 2024) for AMC. Both outperform NICE. Notably, even a small model like Qwen 2.5-3B-Instruct performs better due to its better alignment training, despite its smaller size. These models offer comparable performance to GPT-4 without incurring the additional API cost.

*Table 12.* Performance on the RLHF task using different models for NICE_AMC. Qwen models offer competitive performance without the API cost of GPT-4.

| Model | NICE | NICE AMC (GPT-4) | NICE AMC (Qwen2.5 7B) | NICE AMC (Qwen2.5 3B) |
|---|---|---|---|---|
| RLHF | $2.82_{\pm 0.10}$ | $3.03_{\pm 0.02}$ | $3.00_{\pm 0.03}$ | $2.97_{\pm 0.03}$ |

## I. Ablation Studies

Unless otherwise specified, the ablation studies in the section below are conducted on the Llama2-7B model.

## I.1. Using an Alternative Way to Compute Policy Gradient

*Table 13.* Performance comparison between loss-based influence estimation (LESS) and NICE which uses different approaches to compute policy gradient when selecting from a task-aware training data pool. *PG* refers to using the MC policy gradient as described in Sec. 3.1, while *PPO* denotes using Proximal Policy Optimization. Employing policy gradient computed from either policy optimization approach during data selection results in a generally better-selected subset compared to loss-based influence estimation.

| dataset | RLHF | HumanEval | | |
|---|---|---|---|---|
| metrics | reward | pass@1 | pass@10 | pass@100 |
| LESS | $1.62_{\pm0.05}$ | $13.55_{\pm0.28}$ | $30.53_{\pm0.57}$ | $52.67_{\pm0.71}$ |
| PG | $1.69_{\pm0.05}$ | $13.43_{\pm0.33}$ | $31.70_{\pm0.66}$ | $55.09_{\pm1.66}$ |
| PPO | $1.73_{\pm0.02}$ | $13.58_{\pm0.95}$ | $30.84_{\pm1.52}$ | $52.08_{\pm1.31}$ |

Besides the vanilla Monte-Carlo policy gradient, we conducted an ablation study of using another policy optimization methodology to compute the policy gradient used in NICE. We tried computing the policy gradient using Proximal Policy Optimization (PPO) (Schulman et al., 2017), with results presented in Tab. 13 for the task-aware setting. These results demonstrate that compared to the loss-based influence estimation (LESS), which selects training data by optimizing in the direction of decreasing validation loss, integrating either the MC policy gradient or PPO gradient to select training data by optimizing in the direction of improving downstream task performance leads to a better-selected subset. A "better-selected subset" refers to a subset of training data that, when used for model training, results in improved performance on downstream tasks.

## I.2. Introducing Additional knowledge on Loss-based Influence Estimation

Tab. 14 shows the performance of LESS using GPT-generated labels (LESS+GPT), which is generally worse than our approaches and can even be worse than LESS + true labels (LESS). Hence, simply using GPT-generated labels with loss-based approaches cannot always address the unavailability of labels for validation data.

*Table 14.* Performance of LESS using GPT-generated labels (2nd row) in the task-agnostic setting, which is generally worse than our approaches and can even be worse than LESS + true labels (1st row).

| Table D | Alpaca | TLDR | RLHF | HumanEval |
|---|---|---|---|---|
| LESS | $26.94_{\pm2.37}$ | $3.37_{\pm0.78}$ | $1.44_{\pm0.07}$ | $47.50_{\pm1.57}$ |
| LESS+GPT | $27.35_{\pm1.86}$ | $3.41_{\pm0.26}$ | $3.03_{\pm0.01}$ | $43.04_{\pm1.39}$ |
| NICE | $27.61_{\pm2.12}$ | $3.61_{\pm0.78}$ | $2.82_{\pm0.10}$ | $48.59_{\pm2.08}$ |
| $\text{NICE}_{\text{AMC}}$ | $30.45_{\pm2.39}$ | $3.55_{\pm0.40}$ | $3.03_{\pm0.02}$ | $45.10_{\pm2.84}$ |

## I.3. Effect of reward score

We conduct a simple ablation study to evaluate the effectiveness of the reward score within the context of the policy gradient methodology. For the RLHF task under a task-agnostic training data pool, we compare the performance between the continuous reward score (our current methodology) and a discrete reward score where the reward is set to 1 if positive and 0 if non-positive. This discrete reward setup can also be interpreted as a form of rejection sampling. The subset selected using the continuous reward yields a model with a performance score of $2.82_{\pm0.10}$, whereas the subset selected using the discrete reward yields a performance of $2.25_{\pm0.12}$. Although less effective than the original policy gradient method, the policy gradient with a discrete reward still demonstrates some effectiveness, outperforming both LESS ($1.44_{\pm0.07}$) and random ($2.05_{\pm0.11}$) selection strategies. This ablation study highlights that subsets selected with the guidance of reward scores can produce models with better performance than those trained on subsets selected by LESS or random sampling.

# J. Qualitative analysis

## J.1. Generated Responses

We present three generated responses from GPT-4 on an AlpacaEval validation data point in Tab. 15 and Tab. 16, demonstrating that by using a better-performing model, we can gather responses of comparable or even higher quality than the original label. This supports the use of the better-performing model during data selection to guide further performance improvements.

By comparing the original labels with the GPT-4 responses, particularly in Tab. 15, we observe that the generated responses exhibit higher quality and more accurate results. By observing Tab. 16, we find that the generated responses are not only better in quality, provide richer detail, and more vivid imagery for monologues, but they also offer diverse responses. These monologues differ in style and narrative focus, ranging from a grand epic tone to reflective introspection. The first explores ancient ruins and lost civilizations, the second emphasizes nature and mysticism, and the last tells mythic tales.

### J.2. Highest/Lowest Scoring Training Data for RLHF Task under a Task-agnostic Training Data Pool

In a task-agnostic setting, we present the highest- and lowest-scoring training data points in Tab. 17. The highest-scoring training data points complete the task effectively and offer additional knowledge, while the lowest-scoring ones tend to be vague, minimal, or fail to address the task meaningfully. Considering the RLHF task targets helpfulness, the high-scoring data points can potentially boost the model's helpfulness. These qualitative examples show that NICE can effectively identify more relevant training data by assigning them higher scores.

### J.3. Highest/Lowest Scoring Training Data for RLHF Task under a Task-aware Training Data Pool

In a task-aware setting, we present the highest- and lowest-scoring training data points in Tab. 18. The lowest-scoring training data points whose last-turn responses are either unhelpful or fail to provide additional meaningful information, potentially degrading model performance (we only use the last-turn response during training, as explained in App. A.1). Conversely, the top-scoring examples offer useful responses and can enrich the model's knowledge. Overall, NICE effectively filters out noisy data while favoring high-quality training examples.

*Table 15.* We compare three GPT-4-generated responses with the original labels of an AlpacaEval validation data point, observing that GPT-4 consistently produces higher-quality and more accurate outputs.

| |
|---|
| **Question:** "If the endpoints of a line segment are (2, -2) and (10, 4), what is the length of the segment?" |
| **Baseline Response:** The length of the line segment is 12 units (generated by text_davinci_003). |
| **Generated Response 1:** To find the length of a line segment given its endpoints, you can use the distance formula. The distance formula between two points $(x_1, y_1)$ and $(x_2, y_2)$ in the coordinate plane is: $$d = \sqrt{(x_2 - x_1)^2 + (y_2 - y_1)^2}$$ For the points given, $(x_1, y_1) = (2, -2)$ and $(x_2, y_2) = (10, 4)$. Plugging these into the formula: $$d = \sqrt{(10 - 2)^2 + (4 - (-2))^2}$$ $$d = \sqrt{8^2 + 6^2}$$ $$d = \sqrt{64 + 36}$$ $$d = \sqrt{100}$$ $$d = 10$$ Therefore, the length of the line segment is 10 units. |
| **Generated Response 2:** To find the length of the line segment with endpoints (2, -2) and (10, 4), we use the distance formula. The distance $d$ between two points $(x_1, y_1)$ and $(x_2, y_2)$ is given by: $$d = \sqrt{(x_2 - x_1)^2 + (y_2 - y_1)^2}$$ In this case, the coordinates of the endpoints are $(x_1, y_1) = (2, -2)$ and $(x_2, y_2) = (10, 4)$. Substitute these values into the formula: $$x_2 - x_1 = 10 - 2 = 8$$ $$y_2 - y_1 = 4 - (-2) = 4 + 2 = 6$$ Now substitute these differences into the distance formula: $$d = \sqrt{8^2 + 6^2} = \sqrt{64 + 36} = \sqrt{100}$$ So, the length of the segment is: $$d = 10$$ Therefore, the length of the line segment is 10 units. |
| **Generated Response 3:** To find the length of a line segment with endpoints $(x_1, y_1)$ and $(x_2, y_2)$, we can use the distance formula, which is derived from the Pythagorean theorem. The formula is: $$d = \sqrt{(x_2 - x_1)^2 + (y_2 - y_1)^2}$$ Given the endpoints of the line segment are $(2, -2)$ and $(10, 4)$, we substitute these coordinates into the formula: $$x_1 = 2, y - 1 = -2, x_2 = 10, y_2 = 4$$ . Now, plug these values into the formula: $$d = \sqrt{(10 - 2)^2 + (4 - (-2))^2}$$ $$d = \sqrt{8^2 + 6^2}$$ $$d = \sqrt{64 + 36}$$ $$d = \sqrt{100}$$ $$d = 10$$ Thus, the length of the segment is 10 units. |

*Table 16.* We present three diverse and vividly detailed generated monologues that answer an AlpacaEval validation question, demonstrating how GPT-4 can vary tone, style, and narrative focus in generated responses.

| |
|---|
| **Question:** "Write a monologue. Do not offer any explanation or out of character notes. The setting is in the Elder Scrolls universe." |
| **Baseline Response:** "Ah, what a glorious day! I can feel the energy of the land, the power of the gods, and the strength of the people of Tamriel. I can sense the passion of the Nords, the determination of the Dunmer, the inventiveness of the Argonians, and the wisdom of the Altmer. The air is filled with the sounds of battle, adventure, and hope. I am so proud to be part of this world, to be a part of the Elder Scrolls. I have seen many things in my travels, but nothing compares to the freedom and beauty of this land. I am so blessed to be able to share in its secrets and experience its wonders. I can only hope that I can do something to help protect it and ensure its future." |
| **Generated Response 1:** "Ah, listen, traveler, to the whispers of this old, timeworn place. What stories the stones of the Ayleid ruins might tell, if only they could speak! You see, each crumbling arch and broken pathway, each shadow that dances upon these ancient walls, carries the weight of ages long past, the echoes of a once-mighty civilization now lost to time and tide.

I have walked these lands, from the high, snowy peaks of Skyrim to the lush, vibrant shores of Summerset Isle, and everywhere in between. But it is here, amidst the haunting beauty of these ruins, that I feel the pulse of the eternal, the ceaseless march of history that cares not for the names of kings or the wars of men and mer.

Once, these halls rang with the laughter of the Ayleid, the Heartland High Elves, masters of starlight and stone, whose magic was said to weave through the very air like threads of gold. They built their cities as high as their ambitions, soaring and majestic. But pride, ah, pride—it festers, and it corrupts. It led them to cruel dominion over humans, their slaves, who toiled under an unyielding yoke. The Ayleids' light dimmed with their mercy, and so their empire fell, as all tyrannies must, to rebellion and the relentless tides of change. |
| **Generated Response 2:** "Ah, you find me at a peculiar time, friend, beneath the boughs of Eldergleam, where dreams and reality merge like the river waters at the birthplace of the world. I am but a humble servant of Kynareth, tending to the whispers and the weeping of the wind through these ancient branches. You seek wisdom, or perhaps power? Both are dangerous, more so than the sharpest blade or the darkest incantation.

There was a time, not long ago in the eyes of the gods but perhaps many lifetimes in the minds of men, when I wandered these lands in search of the same. From the chilled breath of Skyrim to the sun-kissed shores of Hammerfell, I walked. I have slept under the stars that wheel unchanging over the Imperial City, and I've faced the dragons' fury, their breath the fire of a thousand forges.

But listen well, for here is the heart of what I have learned: Tamriel is a land of relentless change, of cycles that turn as surely as the moons dance around Nirn. Empires rise and fall, the swords of kings grow dull with rust, and even the mightiest spells fade into echoes. Yet, the trees, the stones, and the rivers—they remember. They hold the long memory of the earth, a memory that outlasts the fleeting reigns of mortals.

Do you feel the roots of the Eldergleam? Do you hear their slow, ceaseless whisper? They speak of the Green Pact and the oaths sworn by the first Bosmer, of the Red Mountain where fire married sky, of the Snow Elves driven deep into the earth's dark embrace. They speak of the rise of heroes and the fall of villains, all threads in the grand tapestry woven by the Aedra and Daedra, each a story bound in the fabric of this world. |
| **Generated Response 3:** "Ah, traveler! Welcome to the warmth of my hearth. Rest your bones by the fire and let the chill of the Skyrim winds be but a distant memory. You must have many tales of your own, but spare an old Mer a moment to regale you with a story of the ancient days, a tale passed down through generations to keep alive the spirit of our people.

We begin in the days when the world was young, when the Aedra walked the land and the Daedra toyed with the threads of fate from their dark realms. In the heart of what you now know as the Rift, there stood an impenetrable forest older than the snows atop the Throat of the World. And within this forest, hidden by the veils of time and enchantment, was the city of Falinesti, the wandering city, a marvel of the Elder Scrolls, moving as if alive through the wilds of Tamriel.

Imagine, oh weary traveler, a city on the limbs of the great Elden Tree, roots deep in the earth, branches scraping the heavens. It was there, in that moving city, that the tale of Aelarion was born. Aelarion, son of no one, a child of the shadows who could speak with the birds and run as swiftly as the West Wind.

Now, Aelarion was a creature of curiosity and mischief. He would watch the traders come and go, telling tales of distant lands, of the deserts of Hammerfell and the snows of Skyrim, where you hail from. He became enamored with the tales of the Dragon Priests, those mighty and fearsome rulers of old. Enthralled by the power they wielded through their masks, relics gifted by dragons, he dreamt of nothing but to behold such a mask himself. |

*Table 17.* We list the training data points with the highest or lowest NICE score for the RLHF task under a task-agnostic training data pool, illustrating that highest-scoring responses are helpful and can convey additional knowledge. By identifying and promoting these high-scoring samples, NICE-selected subset have the potential to enhance the model's overall helpfulness.

| Dataset | Dialogue |
|---|---|
| | Training data points with the highest NICE score |
| Flan V2 | **User:** Detailed Instructions: In this task, you're given an ambiguous question (which can be answered in more than one way). Your task is to provide one question which clarifies the input question and it has one unique answer, and also provide an answer to the generated question. Generated question and answer should be separated with a new line. Problem: When was free health care introduced in Canada? Solution:
**Assistant:** When was nationalized public health care introduced in Canada?
1947 |
| Flan V2 | **User:** Q: In this task, you're given an ambiguous question (which can be answered in more than one way). Your task is to provide one question which clarifies the input question and it has one unique answer, and also provide an answer to the generated question. Generated question and answer should be separated with a new line.
Who got the biggest NBA contract right now?
**Assistant:** Who has the biggest NBA contract salary in the 2016-2017 season?
LeBron James |
| DOLLY | **User:** Task: What team did Bobby Orr play for in his 5th season?
Input: Robert Gordon Orr OC (born March 20, 1948) is a Canadian former professional ice hockey player, widely acknowledged as one of the greatest of all time. Orr used his ice skating speed, scoring, and play-making abilities to revolutionize the position of defenceman. He played in the National Hockey League (NHL) for 12 seasons, the first 10 with the Boston Bruins, followed by two with the Chicago Black Hawks. Orr remains the only defenceman to have won the league scoring title with two Art Ross Trophies. He holds the record for most points and assists in a single season by a defenceman. Orr won a record eight consecutive Norris Trophies as the NHL's best defenceman and three consecutive Hart Trophies as the league's most valuable player (MVP). Orr was inducted into the Hockey Hall of Fame in 1979 at age 31, the youngest to be inducted at that time. In 2017, Orr was named by the National Hockey League as one of the "100 Greatest NHL Players" in history.
Output:
**Assistant:** Bobby Orr played for the Boston Bruins in his 5th season in the National Hockey League (NHL). He played in the NHL for 12 seasons, the first 10 with the Boston Bruins, followed by two with the Chicago Black Hawks. |
| | Training data points with the lowest NICE score |
| Flan V2 | **User:** Translate " ." to Finnish?
**Assistant:** . |
| DOLLY | **User:** Is tap water safe to drink in Japan?
**Assistant:** Yes, tap water is safe to drink in Japan. |
| OASST | **User:** I have trouble staying on track when I need to do schoolwork. What are some ways that I can keep myself on track?
**Assistant:** Getting rid of anything that distracts you could be helpful. |

*Table 18.* We show the training data points with the highest or lowest NICE score for the RLHF task under a task-aware training data pool, demonstrating that top-scoring samples enrich model knowledge while low-scoring ones can be noizy. This filtering capability helps NICE retain data likely to improve downstream task performance.

| Training data points with the highest NICE score |
|---|
| **User:** Mustard seems pretty common these days, in most people's refrigerators. When did people start using mustard as a condiment on foods, I'm curious about its history.
**Assistant:** I don't really know about mustard, so I found an article that you might like, in which the author writes, "Mustard is known to have originated in India but was long ago used as a condiment in ancient China and Egypt." Here's a quote from that article that sounds relevant: "It appears that mustard was used as an ingredient in cooking in several ancient civilizations, including those of India, China, and Egypt. In fact, the oldest known picture of mustard is a carving of a jar on the wall of a tomb in ancient Egypt. This artifact, which dates to the 15th century B.C., shows the Egyptians enjoying a dish that may well have included mustard as a condiment." I'd be happy to read that article for you if you want. |
| **User:** What are some good stretches I should do before going to the gym?
**Assistant:** Stretches are an important part of a workout regimen. They get the blood flowing through the body and help you limber up for movement, which is essential for doing yoga. What types of stretching are you looking to do?
**User:** I was hoping you could tell me.
**Assistant:** Well, before every workout you should do some dynamic stretches, which will bring circulation to the muscles, and some static stretches, which will make the muscles more flexible and less susceptible to injury.
**User:** Ok, which stretches specifically?
**Assistant:** Ok, what about calf raises? |
| **User:** Can you teach me how to make candles?
**Assistant:** Sure, just follow these directions and it should be easy! |
| Training data points with the lowest NICE score |
| **User:** How fast can trains go?
**Assistant:** That depends on the kind of train, the length of the train, the density of the track, the grade, and the direction of the train.
**User:** What about coal trains?
**Assistant:** I'm not sure what you mean by "coal trains." |
| **User:** What's the best brand of TV?
**Assistant:** I'm not sure what you mean by "best brand." |
| **User:** How can I make my clothes whiter?
**Assistant:** You can use vinegar to "bleach" the clothes.
**User:** Vinegar can bleach clothes?
**Assistant:** Yes. |

