# OpenReview forum: "NICE Data Selection for Instruction Tuning in LLMs with Non-differentiable Evaluation Metric"
_ICML.cc/2025/Conference — ICML 2025 poster_

### Official Review · Reviewer_pZkV · 2025-03-12

**Overall Recommendation:** 3

**Summary:**

The paper proposes a data selection framework that computes the evaluation metric (e.g., reward) for validation samples and employs the Monte-Carlo policy gradient to calculate the influence of each training data point on the validation data.

**Claims And Evidence:**

See Other Strengths And Weaknesses

**Essential References Not Discussed:**

See Other Strengths And Weaknesses

**Experimental Designs Or Analyses:**

See Other Strengths And Weaknesses

**Methods And Evaluation Criteria:**

See Other Strengths And Weaknesses

**Other Comments Or Suggestions:**

See Other Strengths And Weaknesses

**Other Strengths And Weaknesses:**

### Strengths
1. The issue of data efficiency in instruction tuning datasets is an important topic in scaling LLMs.
2. The paper is easy to understand


### Weaknesses
1.  The claim about validation loss vs. evaluation metrics may be imprecise. The authors primarily base their argument on a single experiment with a reward model (Figure 1). This raises concerns that the relationship between validation loss and evaluation metrics may not be generalizable and cause misunderstandings.

2. The motivation seems unclear.

(1) The paper argues that high-quality labels are crucial yet often unavailable for loss-based data selection methods. It then relies on an additional model (e.g., a reward model or LLM judge) to generate the evaluation metric. How do the authors ensure the quality and reliability of these metrics?

(2) There is insufficient evidence in the paper showing that the lack of labels is the main limitation of previous loss-based data selection methods. How will existing state-of-the-art loss-based data selection methods perform if established LLMs or [1] can be used to generate validation labels or according to the teacher-forcing loss?

3. Insufficient experiments to support the claim about the advantage of policy gradient compared to the gradient used in previous loss-based methods. Note that previous gradient-based methods might also be improved by incorporating multiple label responses from additional LLMs. Thus, it remains unclear whether the benefit arises from Monte Carlo sampling or from the introduced evaluation metric itself.

4. Some technical details remain unclear. For example, the paper projects the LoRA gradient into an 8192-dimensional vector in Line 290, but it does not illustrate the projection method used.

5. The choices of baselines may be outdated and less than ideal.

(1) The paper introduces additional models to help data selection, which may create an unfair comparison against other baselines.

(2) Some baselines (e.g., LESS) are specifically designed for target data selection, so applying them in a task-agnostic context may be inappropriate. I suggest the author further compare and discuss more recent data selection methods in each setting.

6. Monte Carlo sampling can be expensive, and the paper does not offer a comparative efficiency analysis of the computational cost for the proposed method and other baselines.

[1] The ALCHEmist: Automated Labeling 500x CHEaper than LLM Data Annotators. In NIPS, 2024

## update after rebuttal
I appreciate that most of my concerns have been addressed. Consequently, I have increased my score to 3. I expect the authors to incorporate new results, analyses, and clarifications in the revised manuscript.

**Questions For Authors:**

See Other Strengths And Weaknesses

**Relation To Broader Scientific Literature:**

See Other Strengths And Weaknesses

**Theoretical Claims:**

There are no theoretical claims.

---

> ### Author Rebuttal · Authors · 2025-04-01
>
> We thank the reviewer for highlighting the importance of our work on data efficiency in instruction tuning and for finding the paper easy to understand. Below are clarifications to address your concerns:
> 1. Since discrepancy between NTP loss & eval. metrics has been shown in previous works (lines 380-1 L for refs.), it explains why we only consider a single experiment to further validate this claim. Nevertheless, we conduct experiments on 3 other datasets with different reward functions (eval. metrics) to show that the discrepancy between loss & metrics can be generalized. The results in [link](https://postimg.cc/GBMkCw0q) are similar to that in Fig.1: checkpoints with minimal loss (highest negative loss) do not correspond to checkpoints with the best performance (measured by the eval. metric) (lines 394-6 L); the performance can continue to increase even if the negative loss decreases.
> 2. Our work can be motivated using two limitations of loss-based data selection: (a) discrepancy between NTP loss & eval. metrics as discussed in point 1 (lines 19-43 R) & (b) dependency on high-quality labels which can be unavailable in practice (lines 149-51 L). In contrast, NICE can select training data to directly optimize commonly used eval. metrics of generation tasks (lines 46-9 R) & alleviate the dependency on validation labels when the eval. metric is independent of labels (lines 64-8 L).
>
> 2.1. High-quality labels (ground-truth responses) are often unavailable or available only in small quantities for generation tasks. So, a common practice is to use an LLM judge or train a reward model based on the small quantity of high-quality labels, and assume they are reliable, high-quality metrics that generalize to other inputs. Thus, this does not contradict the importance of high-quality labels. In fact, prior works have shown that these eval. metrics can track the response quality & align well with humans (Bai et al. 2022; Dubois et al., 2024; Stiennon et al., 2020; Zheng et al., 2023) and are hence widely adopted for evaluating generation quality.
>
> 2.2. Tab.D below shows the performance of LESS using GPT-generated labels (as suggested, 2nd row), which is generally worse than LESS + true labels (1st row) and our approaches. Hence, the performance of loss-based data selection is hurt by the unavailability of true labels, highlighting their importance. Furthermore, using GPT-generated labels alone does not address the issue discussed in point 1.
> |Table D|Alpaca|TLDR|RLHF|pass@1|pass@10|pass@100|
> |-|-|-|-|-|-|-|
> |LESS|15.44±2.74|1.54±0.21|1.44±0.07|0.092±0.01|0.261±0.00|0.475±0.02|
> |LESS+GPT|15.07±0.91|1.47±0.60|3.03±0.01|0.043|0.188|0.417|
> |NICE|16.79±0.85|1.61±0.39|2.82±0.10|0.104±0.02|0.274±0.01|0.486±0.02|
> |NICE AMC|17.16±3.73|3.55±0.40|3.03±0.02|0.090±0.02|0.251±0.02|0.451±0.03|
>
> 3. Advantage of NICE (policy gradients) over LESS (loss-based gradients) is supported by experimental evidence in Tab.3: NICE generally outperforms LESS (lines 321-9 L). About the concern on MC sampling & the eval. metric $r$, note that the policy gradient is used to optimize $r$ with data selection and approximated by MC sampling. So, it won’t be meaningful if we consider MC sampling as a standalone component and remove the eval. metric (e.g., for an ablation study). To see this, not using the eval. metric is equivalent to treating all sampled responses equally, i.e., $r$ is a constant function, resulting in a policy gradient of 0, as derived in [link](https://postimg.cc/5jGSyF71).
>
> 4. We use Johnson-Lindenstrauss random projections (see lines 275-8 L).
>
> 5.1. In Tab.D above, we have provided the results of LESS + GPT-generated labels, making the comparison fair. However, as mentioned in point 2.2, LESS is using the true labels to outperform that with GPT-generated labels and thus has an unfair advantage (yet poorer performance) over our approach in TLDR, RLHF & HumanEval (lines 278-81 R). Moreover, NICE (without AMC) already outperforms LESS without the help of additional models (Tab.3).
>
> 5.2. To clarify, our “task-agnostic” & “task-specific” settings pertain to the data preparation stage (to form the initial pool of training data without & with the knowledge of the target task, resp.) **prior** to data selection (lines 245-8 R, 255-7 R). For data selection, this work focuses on targeted (task-specific) data selection (lines 102-5 R). Hence, all our baseline choices are fair and valid. We will clarify this in the revised paper. In addition, we have included 2 more baselines TSDS & DSIR in Tab.C of reviewer d8As’s rebuttal.
>
> 6. While NICE incurs more computational cost due to MC sampling, this trade-off is justified by not needing validation labels—a key motivation of this work—and its improved performance over other methods (Tab.3). We defer the computational analysis to reviewer waTE’s rebuttal.
>
> We will include the above discussions in the revised paper. We hope we have addressed your concerns and improved your opinion of our work.

---

> > ### Comment · Reviewer_pZkV · 2025-04-09
> >
> > I appreciate the authors' efforts to address most of my concerns. However, after reviewing all comments and the rebuttal, I still tend to maintain my score due to the following concerns:
> >
> > The claims about this paper's “task-agnostic” & “task-specific” settings are inconsistent and might contradict existing work.  The author claims that the “task-agnostic and task-specific settings pertain to the data preparation stage prior to data selection”. However, they later compared data selection methods based on task-agnostic and task-specific categories.  Also, the explanations provided in both the paper (Lines 245–248 and 255–257) remain unclear. For example, LESS is described as a task-specific data selection method, but it utilizes mixed-source training datasets that are not collected specifically for the downstream evaluation task. In Table 3, some baselines like LESS are designed as target data selection methods, suggesting that applying them in a task-agnostic context could also be inappropriate.
> >
> > I am still concerned that these inconsistencies may lead to misunderstandings within the community.

---

> > > ### Author Response · Authors · 2025-04-09
> > >
> > > We thank the reviewer for reviewing our rebuttal and for allowing us to further clarify your concerns.
> > >
> > > **Problem setting:** We only focus on targeted data selection where we assume we have access to some validation data during data selection (lines 100-105, Left). Our “task-agnostic” and “task-specific” are used to describe the property of the **initial pool of training data** collected in the **data preparation** stage **prior** to data selection. Specifically, for the data preparation stage:
> > >
> > > - For the "task-agnostic" setting: the initial pool of training data is gathered **without** any knowledge of the target task. We used a mixed-source instruction-tuning training set in this setting. Intuitively, this mixed-source pool of data may contain many irrelevant data (e.g., assistant-style conversations) w.r.t. the targeted task (e.g., coding task).
> > > - For the "task-specific" setting: the initial pool of training data is gathered **with** an explicit focus on the target task. We select from the training data that mainly contains data relevant to the target task, e.g., selecting from codes (CodeAlpaca 20k) for a target coding task (HumanEval).
> > >
> > > **Perceived inconsistencies:** In the main paper, we did **not** explicitly categorize data selection methods using the terms “task-agnostic” and “task-specific” (even though they can be categorized in this way). As we focus on targeted data selection, we only and fairly compare methods that require the validation datasets and are applicable for **targeted** data selection. In addition, as our “task-agnostic” and “task-specific” settings refer to the property of the initial training data pool (**not** the availability of the validation data during the selection stage), all targeted data selection methods can be **appropriately** applied to these two types of pool of data without inconsistency.
> > >
> > > **Perceived contradictions with existing work**: We would also like to clarify that our settings do not contradict existing work. Our methodology aligns with established practices, such as those seen in previous works like LESS [1], which focus on "task-agnostic" data preparation, followed by targeted data selection. Specifically, our “task-agnostic” setting's mixed-source training dataset is the same training data used in LESS. In addition, we **extend** their experimental setup by including a "task-specific" (data preparation) setting, demonstrating the applicability of our approach (NICE) to more scenarios without contradicting the settings observed in prior studies.
> > >
> > > We will improve the clarity of the terms in the revised version. We hope that we have clarified that there is NO contradiction or inconsistency. If the confusion lies in the naming conventions, we can update “task-specific” to “task-aware” for the data preparation stage, which does not compromise the contributions of this work.
> > >
> > > [1] Xia, Mengzhou, et al. "Less: Selecting influential data for targeted instruction tuning." *arXiv preprint arXiv:2402.04333* (2024).

---

### Official Review · Reviewer_d8As · 2025-03-14

**Overall Recommendation:** 4

**Summary:**

The paper proposes NICE, a RL based framework for choosing instruction tuning data targeted for downstream tasks. The proposed method uses reward signals such as loss function or influence function on the validation data. The policy gradient is then used to estimate the training data point's influence on the given validation sets. The authors conducted experiments on instruction tuning of LLMs such as llama and mistral models.

**Claims And Evidence:**

The authors claims their selection strategy is better than loss based strategy and other existing data selection strategies. There has also been other recent literature in this space such as https://arxiv.org/abs/2410.11303. The authors should compare to the methods proposed in this paper.

The authors also claim their method works well in task agnostic settings, but fail to compare against some of the existing literature in this space, such as the methods that use DPP (https://arxiv.org/pdf/2402.02318) and submodular optimization (https://arxiv.org/abs/2401.06692).

**Essential References Not Discussed:**

In addition to the papers mentioned above, the authors are also encouraged to check out other methods mentioned in https://arxiv.org/pdf/2402.16827.

**Experimental Designs Or Analyses:**

I did not find the specific details when the labels are unavailable. I think the paper could benefit from a better description what what was used in place of the labels $y$ in the experiment section.

**Methods And Evaluation Criteria:**

The methods and evaluation make sense and are quite thorough other than the baselines they compare against.

**Other Comments Or Suggestions:**

N/A

**Other Strengths And Weaknesses:**

Overall, the paper is very well written. I am willing to raise my score given the experiment comparisons with other baselines are conducted.

**Questions For Authors:**

N/A

**Relation To Broader Scientific Literature:**

The paper falls under the general data selection for LLM instruction tuning.

**Theoretical Claims:**

I did check the theoretical derivations.

---

> ### Author Rebuttal · Authors · 2025-04-01
>
> We thank the reviewer for the positive feedback on our writing and the thoroughness of our methods and evaluations. We wish to make the following clarification.
>
> **Claims And Evidence:**
>
> To clarify, our “task-agnostic” & “task-specific” settings pertain to the data preparation stage (to form the initial pool of training data without & with the knowledge of the target task, resp.) **prior** to data selection (lines 245-8 R, 255-7 R). For data selection, this work only focuses on targeted (task-specific) data selection (lines 102-5 R).
>
> However, methods based on DPP [https://arxiv.org/pdf/2402.02318] and submodular optimization [https://arxiv.org/abs/2401.06692] focus on non-targeted (task-agnostic) data selection without validation data, which is a different problem setting that falls outside the scope of our work. We will improve the clarity in the revised paper.
>
> To improve the comprehensiveness of the experiments for targeted data selection, we add TSDS [https://arxiv.org/abs/2410.11303] (as suggested) and DSIR (highlighted in the suggested survey [https://arxiv.org/pdf/2402.16827] [1] ) for the setting where the initial pool of training data does not have the knowledge of the target task (our "task-agnostic" setting). The results are shown in Tab. C below.
>
> Across multiple datasets and models, **our method generally continues to outperform the newly added baselines**. Although DSIR performs the best on AlpacaEval with the LLaMA2-7B model, it selects data based on n-gram lexical feature matching (similar to BM25) between training and target distributions, independent of model-specific signals. As a result, it selects the same subset regardless of the model used for instruction tuning, which can be sub-optimal. We can see it is no longer the best with Mistral-7B on the same task. This highlights the advantage of our model-aware data selection strategy. We will incorporate these baselines and citations in the revised paper.
>
> |Table C|Alpaca|TLDR|RLHF|pass@1|pass@10|pass@100|
> |-|-|-|-|-|-|-|
> |**Llama**|
> |TSDS|11.42±1.33|1.42±0.07|1.01±0.12|0.1030±0.02|0.2547±0.01|0.4368±0.02|
> |DSIR|20.27|1.53|2.57|0.0953|0.2402|0.4222|
> |NICE|16.79±0.85|1.61±0.39|2.82±0.10|0.1035±0.02|0.2737±0.01|0.4859±0.02|
> |NICE AMC|17.16±3.73|3.55±0.40|3.03±0.02|0.0904±0.02|0.2511±0.02|0.4510±0.03|
> |**Mistral**|
> |TSDS|25.91±2.49|3.47±0.00|1.83±0.15|0.2750±0.01|0.5978±0.02|0.8278±0.01|
> |DSIR|29.31|3.48|2.94|0.2771|0.5681|0.7917|
> |NICE|26.20±4.13|3.31±0.35|3.10±0.06|0.2948±0.01|0.6205±0.02|0.8559±0.01|
> |NICE AMC|31.05±1.23|4.60±0.20|3.42±0.05|0.2996±0.02|0.6210±0.02|0.8567±0.00|
>
> **Experimental Designs Or Analyses:**
>
> We would like to clarify that for three tasks—TLDR, RLHF, and HumanEval—the ground-truth labels of the validation dataset are not used by our method, although they are available to the baselines (lines 305–18, L). For AlpacaEval, labels are provided to all approaches. Specifically, NICE relies solely on the probability of generated responses and the score from a reward function, as enabled by the policy gradient mechanism (lines 130–132, R). In these three tasks, the **reward function does not require ground-truth labels**. Specifically:
>
> - For RLHF and TLDR, the reward function is a learned reward model that outputs scores based on the prompt and generated response (lines 160-162, L).
> - For HumanEval, the reward is defined by whether the generated code passes unit tests, not requiring reference solutions (lines 245-250, L).
>
> We will make this distinction clearer in the experiment section to avoid confusion.
>
> **Essential References Not Discussed:**
>
> We have checked the survey paper in https://arxiv.org/pdf/2402.16827 and added a related work DSIR as a baseline for comparison. We will incorporate the citations of the survey paper and works that are not directly comparable but related like DPP, SKILL-IT[2] in the related work section.
>
> We thank the reviewer for the valuable comments. We hope the additional baselines make our comparison more thorough and improve your impression of our work.
>
> [1] Xie, Sang Michael, et al. "Data selection for language models via importance resampling." *Advances in Neural Information Processing Systems* 36 (2023): 34201-34227.
>
> [2] Chen, Mayee, et al. "Skill-it! a data-driven skills framework for understanding and training language models." Advances in Neural Information Processing Systems 36 (2023): 36000-36040.

---

> > ### Comment · Reviewer_d8As · 2025-04-03
> >
> > Could you please clarify how the validation data (for the targeted selection) makes a different for task-agnostic settings? Intuitively, the task agnostic setting does seem to benefit much from having separate validation set since the data are all iid?

---

> > > ### Author Response · Authors · 2025-04-05
> > >
> > > We thank the reviewer for the question and for allowing us to clarify.
> > >
> > > Our “task-agnostic“ setting pertains to the **data preparation stage**, where the initial pool of training data is constructed *without any knowledge of the target task or validation data*. This ensures that the process of collecting this initial pool of training data remains entirely task-agnostic. As such, the validation data (for the targeted selection) does not provide any benefit at this stage since it is not used.
> > >
> > > On the other hand, the validation data/set (for the targeted selection) is utilized **during the data selection stage**, where the objective is to identify a subset of training data that maximizes performance on the target task’s validation set and supports the desired target capability (e.g., coding, summarization). For example, NICE selects training examples whose gradients are more similar to the policy gradient of validation data points. Including such examples improves validation performance and thereby supports better performance for the target task.
> > >
> > > Therefore, while the validation data/set **does not make a difference during the task-agnostic data preparation stage (our “task-agnostic” setting)**, it is crucial during the **targeted selection stage** by informing our algorithm to select training data that improve performance for the target task. We hope this clarifies our definition of the “task-agnostic” setting and the role of validation data in our experiments. We also hope it helps to address your concern and can improve your opinion of our work.

---

### Official Review · Reviewer_waTE · 2025-03-14

**Overall Recommendation:** 4

**Summary:**

This paper proposes NICE, which provides an innovative and label-efficient approach to data selection by using policy gradients for non-differentiable evaluation metrics, outperforming several baselines in many benchmarks, including both task-specific and -agnostic settings. While its computational cost and complexity may limit its scalability, it seemly shows promise for improving model performance.

**Claims And Evidence:**

Yes.

**Essential References Not Discussed:**

By following LESS, this paper introduces a data influence selection strategy based on evaluation metrics rather than loss or gradients that can be applied to both task-specific and -agnostic settings. I believe it would be beneficial to incorporate a discussion and experimental performance comparison with [1], even though [1] approaches data selection from the perspective of distribution matching.

[1]TSDS: Data Selection for Task-Specific Model Finetuning

**Experimental Designs Or Analyses:**

Yes.

**Methods And Evaluation Criteria:**

Yes.

**Other Comments Or Suggestions:**

Please refer to the Questions.

**Other Strengths And Weaknesses:**

Strengths:

The authors' insight that there is no direct correlation between the validation set loss and the final test performance is reasonable and profound. Instead, directly using the performance reward for data selection and solving it through reinforcement learning is highly promising and adds innovation to this paper.

Weaknesses:

1. In the paper, RL is used to solve the data selection problem based on performance rewards, which involves computationally expensive steps such as MC sampling. This introduces significant computational cost and instability. Is there a more in-depth discussion on the computational cost and stability? This could affect the reproducibility of the work and its subsequent use in future research.

2. Is there any potential data leakage in the experiments? For example, in Line 258, it is mentioned that the validation set of AlpacaEval and HumanEval share the same distributions as the corresponding test sets. If we were to construct a validation set from a subset of the training dataset without any validation data and without knowledge of the downstream evaluation data, how would the model perform in the task-agnostic setting?

3. There appears to be some inconsistency in the performance between NICE and NICE_AMC in Table 3. How can this be explained? Additionally, could the authors provide some theoretical guidance on when to choose NICE over NICE_AMC or vice versa?

**Questions For Authors:**

Please refer to the Questions.

**Relation To Broader Scientific Literature:**

Please refer to the Questions.

**Theoretical Claims:**

Yes.

---

> ### Author Rebuttal · Authors · 2025-04-01
>
> We thank the reviewer for recognizing our insight and our RL-based data selection approach. Please find our responses below.
>
> **Essential References Not Discussed:**
>
> We have included TSDS (as suggested) and an additional baseline, DSIR in Tab. C of our response to Reviewer d8As. Our method outperforms the new baselines across diverse datasets and models.
>
> **Weaknesses:**
>
> 1.1 **Computation cost**
>
> Although Monte Carlo sampling increases the computational cost, **this trade-off is justified by NICE not needing validation labels** - a key motivation of our work. NICE **fills a gap left by existing loss-based baselines by supporting data selection with unlabeled validation data** in cases where the evaluation metrics are label-independent. Furthermore, we can observe the performance improvement over other methods in Tab.3.
>
> Nevertheless, we provide a comparative analysis of the computational costs between NICE and LESS, showing **NICE remains within a practical computational range:** Tab. A lists the asymptotic complexity and wall-clock runtime (training is measured on a single H100 hour, others are on L40 GPU hours) for each stage in the data selection procedure. Tab. B highlights the validation gradient computation where NICE differs from NICE. Let $E, d, M$ denote number of epochs (saved checkpoints), dimension of the projected gradients, number of MC samples. Let $D_W, D_N, D_V$ denote the warmup, training, validation set. When the sizes of the validation set and the MC samples are small, NICE adds only marginal overhead to LESS (e.g., AlpacaEval).
>
> |Table A|Warmup LoRA|Training Grad Comp|Validation Grad Comp|Data Selection|
> |-|-|-|-|-|
> |Remark|LESS=NICE|LESS=NICE|NICE>LESS|LESS=NICE|
> |Compute|$O(\|D_W\|E)$ 3h|$O(\|D_N\|E)$ 48h|LESS:$O(\|D_V\|E)$ 0.11h on average; NICE: $O(\|D_V\|EM)$ 14.67h on average|$O(\|D_N\|\|D_V\|d)$ < 0.02h|
>
> |Table B| |NICE|NICE|LESS|
> |-|-|-|-|-|
> |$\|D_V\|​$|$M$​|MC Sample|Val Grad|Val Grad|
> |Alpaca=10|20|0.17h|0.05h|<0.02h|
> |TLDR=322|20|8h|1.47h|0.08h|
> |RLHF=2192|20|32h|10h|0.33h|
> |HumanEval=10|500|5h|2h|<0.02h|
>
> 1.2 **Stability**
>
> We provide an ablation study with results in https://postimg.cc/Mf0N6b76, varying MC samples from 5 to 20 on the RLHF dataset for a more in-depth discussion on stability. Results show that increasing the number of MC samples $M$ generally lowers the standard deviation across runs with different seeds, indicating better stability. The benefit of reduced standard deviation diminishes as $M$ increases. This validates that our chosen $M$ provides a good trade-off, offering sufficient stability without excessive computation.
>
> 2. We ensure there is no data leakage in our experimental setup, since the test data has no overlap with the validation data. Additionally, we clarify that similar to LESS and TSDS, our work focuses on targeted data selection (lines 102-5 R), which explicitly assumes access to a validation set that is distributionally aligned with the target task. Our “task-agnostic” & “task-specific” settings in the experiments pertain to the **data preparation** stage (to form the initial pool of training data without & with the knowledge of the target task, resp.) **prior** to data selection (lines 245-8 R, 255-7 R).
>
> We believe that selecting training data without the knowledge of the downstream evaluation data falls under a different setting—task-agnostic data selection—where alternative approaches based on perplexity or coreset are applicable. However, these directions are outside the scope of this work, focusing on targeted data selection.
>
> 3. NICE_AMC generally performs well under two conditions: (a) when a stronger model is available for the target task—one that can generate higher-quality Monte Carlo samples, and (b) when the training pool is sufficiently large, allowing better gradient alignment with NICE_AMC (lines 313–5, R). This explains its success in our task-agnostic (data preparation) setting with abundant data.
>
> As for guidance on when to use NICE vs. NICE_AMC:
>
> - NICE_AMC: a stronger assistance model is available + a large, diverse training pool can be leveraged.
> - NICE: resources or training data are limited or when the base model is already powerful enough.
>
> In the revision, we will include additional baselines and analyses, and clarify the use case of NICE_AMC. We hope the discussion above will address your concerns and improve your impression of our work.

---

> > ### Comment · Reviewer_waTE · 2025-04-06
> >
> > Thanks for the authors' feedback and I don't have other concerns.

---

### Official Review · Reviewer_znd4 · 2025-03-15

**Overall Recommendation:** 4

**Summary:**

The paper introduces NICE (Non-differentiable evaluation metric-based InfluenCe Estimation) for selecting training data to improve the performance of large language models (LLMs) on specific tasks. The method leverages policy gradient techniques to optimize non-differentiable evaluation metrics directly, addressing the limitations of existing loss-based influence estimation methods.

**Claims And Evidence:**

Supported:
1. Effectiveness of NICE: The paper claims that NICE outperforms existing data selection baselines across diverse scenarios.
2.  The authors argue that optimizing non-differentiable evaluation metrics directly leads to better data selection than traditional loss-based methods. The experimental results, particularly in tasks requiring long-form generation, support this claim by demonstrating improved performance metrics.
3. The paper claims that NICE can perform data selection without labeled validation data when the reward function does not require labels.

Problematic:
1. While the paper acknowledges the computational cost of NICE, it claims that the method is efficient due to the use of LoRA and random projection. However, there is a lack of detailed analysis or evidence comparing the computational cost of NICE with other methods LESS or BM25.

**Essential References Not Discussed:**

None.

**Experimental Designs Or Analyses:**

1. The computational cost of Monte Carlo sampling (e.g., 500 samples for HumanEval) and reliance on GPT-4 for AMC are mentioned but not quantified. I also wonder about the cost of use of GPT4 API. Can your method also work well with other open-source or smaller LLMs? You may also discuss trade-offs between sample size and performance.

**Methods And Evaluation Criteria:**

Overall, the proposed methods and evaluation criteria are well-aligned with the problem of optimizing data selection for instruction tuning in LLMs. They address the key challenges and provide a comprehensive framework for evaluating the effectiveness of the proposed approach.

**Other Comments Or Suggestions:**

1. While the paper notes that larger subsets can harm performance (Fig. 3), it does not systematically characterize "harmful" data (e.g., via qualitative examples).

**Other Strengths And Weaknesses:**

Please see the comments in previous sections.

**Questions For Authors:**

Please see the comments in previous sections.

**Relation To Broader Scientific Literature:**

1. NICE advances the field by directly optimizing for task-specific performance metrics, which is a novel approach compared to traditional methods that rely on proxy metrics like next-token prediction loss.
2.  By applying policy gradient techniques to data selection, the paper introduces a new way to estimate the influence of training data on model performance, bridging the gap between reinforcement learning and data selection in NLP.

**Theoretical Claims:**

The theoretical claims in the paper are based on sound principles from reinforcement learning and influence estimation.
I found that the connection between policy gradients and influence estimation is intuitive but lacks formal grounding. The paper does not clarify why policy gradients are theoretically suitable for influence estimation.

---

> ### Author Rebuttal · Authors · 2025-04-01
>
> We thank the reviewer for acknowledging the effectiveness of NICE and the soundness of our methods and evaluation.
>
> **Claims And Evidence:**
>
> We defer the computational analysis between LESS and NICE to our rebuttal for Reviewer waTE. For BM25, it is indeed an efficient retrieval method based on lexical matching, ranking training data by relevance to the validation data. However, **BM25 is model-agnostic** (i.e., it selects the same data for different models). Thus, a subset selected may perform well with one model but not necessarily be optimal with another (lines 286-94, R). While being **model-agnostic contributes to BM25's efficiency, it also limits its performance**. In contrast, NICE enables model-aware data selection, optimizing training data specifically for the target model’s validation performance.
>
> **Theoretical Claims:**
>
> The theoretical suitability of policy gradient for influence estimation can be justified as follows. Loss-based influence estimation methods (e.g., TracIn, influence function), which our method builds upon, estimate the influence of a training point on validation loss. In particular, these methods measure the influence via the “gradient of the validation loss” (the change in validation loss w.r.t. the model parameters / policy), which by the **chain rule**, can be combined with the the first-order gradient or the Hessian of the training loss (w.r.t. the model parameters). In contrast, we estimate the influence on **validation performance**, measured by evaluation metrics. However, since the evaluation metrics are non-differentiable, the policy gradient, which measures the change in validation performance caused by the corresponding training data point, is a direct replacement of the “gradient of the validation loss”. It also allows the estimating the influence of the training data via chain rule.
>
> Moreover, we show elaborate the derivations for NICE (based on TracIn) and NICE_IF (based on influence functions) in lines 141–52 R and lines 201-19 L, respectively. Both of them also relies on the derivations shown in App A.5.1 and App A.5.2, respectively.
>
> Therefore, our NICE is theoretically-grounded since it uses the principled influence estimation framework from prior work and extends it to calculate the influence of data on the non-differentiable metrics using policy gradient.
>
> **Experimental Designs Or Analyses:**
>
> **Cost of GPT-4.**
>
> Note that NICE AMC is an optional enhancement—NICE itself does not require GPT-4. We list the projected GPT-4 cost for NICE AMC in the table below. The costs are low for the majority of the tasks, except for RLHF due to its large validation set (which can be addressed as below).
>
> ||AlpacaEval|TLDR|RLHF|Humaneval|Avg|
> |-|-|-|-|-|-|
> |GPT Cost($)|1.70|14.26|291.17|6.34|78.37|
>
> **Use of open-source/smaller LLMs.**
>
> As suggested, to reduce cost, we can use high-performing open-source models. On the RLHF dataset, we use Qwen 2.5-7B/3B-Instruct for AMC. Both outperform NICE. Notably, even Qwen 2.5-3B performs better due to its better alignment training, despite its smaller size. These models offer comparable performance to GPT-4 without the API cost.
>
> ||NICE|NICE AMC (GPT4)|NICE AMC (Qwen2.5 7B)|NICE AMC (Qwen2.5 3B)|
> |-|-|-|-|-|
> |RLHF|2.82±0.10|3.03±0.02|3.00±0.03|2.97±0.03|
>
> **500 Monte Carlo samples on HumanEval.**
>
> We use 500 samples because HumanEval is challenging—correct generations are rare, and we aim to estimate pass@100, whose evaluation requires 100 generated code pieces. In contrast, other tasks are evaluated using one sample, and we use 20 samples (i.e., a multiplier of 1) to approximate expected performance under the current policy.
>
> **MC sample size vs. performance trade-off.**
>
> We perform experiments on HumanEval with MC sample sizes ranging from 200 to 500 (see Fig. in [link](https://postimg.cc/ZCdzgpBM)), showing a clear performance improvement with more samples. We also analyze this trade-off in Sec. 4.5 (lines 374-84, R). Both HumanEval and RLHF results show performance can improve with sample size. On RLHF, 20 samples already yield strong performance, with additional gains as the size increases.
>
> **Other Comments Or Suggestions:**
>
> We provide qualitative examples in App. A.10 (pages 23–24). These examples illustrate that “harmful” data can include paraphrased versions of the questions or fail to provide useful information.
>
> We will include additional experiments in the revised paper and hope our justifications will address your concerns and improve your opinion of our work.

---

> > ### Comment · Reviewer_znd4 · 2025-04-05
> >
> > Thanks for the response. I found that most of my concerns has been addressed.

---

### Decision · Program_Chairs · 2025-05-01

**Decision:**

Accept (poster)

**Comment:**

The paper proposes NICE for selecting instruction-tuning data by leveraging policy gradients to estimate the influence of training data on validation performance.

The main strengths of the paper lie in the novelty compared to existing data selection approach for instruction tuning, as agreed by all reviewers.

While reviewers acknowledged the method’s novelty and practical utility, concerns centered on computational overheads, baseline comparisons, generalizability. Post-rebuttal, the authors strengthened empirical claims with expanded baselines (e.g., TSDS, DSIR) and computational analysis but left unresolved questions about generalizability.